# Trustworthy Federated Label Distribution Learning under Annotation Quality Disparity

**Junxiang Wu** [* 1 2] **Zhiqiang Kou** [* 1 2] **Hongwei Zeng** [3] **Wenke Huang** [4] **Biao Liu** [1 2]
**Hanlin Gu** [5] **Yuheng Jia** [1 2] **Di Jiang** [6] **Yang Liu** [6] **Xin Geng** [1 2]

## Abstract

Label Distribution Learning (LDL) models supervision as an instance-wise probability distribution, enabling fine-grained learning under inherent ambiguity, but its success relies on high-fidelity label distributions that are costly to obtain and thus often noisy. Motivated by privacy-sensitive applications, we study *Federated Label Distribution Learning* (Fed-LDL), where data isolation further induces heterogeneous annotation quality across clients, making local updates unevenly reliable and breaking sample-size-based aggregation (e.g., FedAvg). To address this trust dilemma, we propose *FedQual*, a quality-aware Fed-LDL framework with two coupled mechanisms: (i) *quality-adaptive client training* guided by a global semantic anchor that calibrates low-quality clients while preserving high-quality autonomy, and (ii) *reliability-aware server aggregation* that reweights client contributions by effective reliable information rather than raw sample size. To enable rigorous evaluation, we construct four new Fed-LDL benchmarks (FER-LDL, FI-LDL, PIPAL-LDL, and KADID-LDL) with controlled annotation quality disparity. We further provide a theoretical guarantee showing that under heterogeneous supervision quality, client-specific calibration is strictly better than any uniform calibration. Experiments on the proposed benchmarks demonstrate the effectiveness of FedQual.

## 1. Introduction

Label Distribution Learning (LDL) (Geng et al., 2007) models supervision as a probability distribution, providing a principled way to capture the inherent ambiguity in visual understanding. Unlike single-label or multi-label classification that enforces a definitive decision, LDL quantitatively describes the degree to which each label is relevant to an instance (Geng, 2016; Kou et al., 2024), and has been widely adopted in medically relevant applications, such as Breast Tumor Cellularity Assessment (Li et al., 2022), Bone Age Assessment (Chen et al., 2022; Zheng et al., 2024), and mental-health analysis (e.g., depression detection) (She et al., 2021; Le et al., 2023). However, the effectiveness of LDL critically depends on the quality of supervision. Constructing reliable label distributions [1] theoretically requires aggregating diverse opinions from a sufficiently large pool of annotators (Kou et al., 2025a; Xu & Zhou, 2017). In practice, such high-fidelity annotation is costly and often infeasible (Xu et al., 2021), resulting in noisy label distributions that force models to learn from ambiguous supervision rather than precise ground truths (Ren et al., 2019).

Given its applications in sensitive domains, the research community has pivoted toward *Federated Label Distribution Learning (Fed-LDL)*. As illustrated in Figure 1(a), this paradigm enables **collaborative training** by keeping raw data on local devices and only transmitting model updates. However, such an architecture significantly exacerbates the previously noted precision bottlenecks. Due to **data isolation** on local devices (Chai et al., 2024) (Liu et al., 2024), the expertise of annotators naturally varies across different clients, resulting in heterogeneous label noise levels. This introduces a unique set of obstacles to the standard Fed-LDL paradigm, which we categorize into **three key challenges**:

Challenge ▲: Annotation Quality Heterogeneity (AQH). As

*Equal contribution [1] School of Computer Science and Engineering, Southeast University, Nanjing, China [2] Key Laboratory of New Generation Artificial Intelligence Technology and Its Interdisciplinary Applications (Southeast University), Ministry of Education, China [3] University of the Chinese Academy of Sciences, China [4] College of Computing and Data Science, Nanyang Technological University, Singapore [5] WeBank, China [6] Academy for Artificial Intelligence, Hong Kong Polytechnic University, Hong Kong, China. Correspondence to: Zhiqiang Kou <zhiqiang_kou@seu.edu.cn>, Yuheng Jia <yhjia@seu.edu.cn>.

*Proceedings of the $43^{rd}$ International Conference on Machine Learning*, Seoul, South Korea. PMLR 306, 2026. Copyright 2026 by the author(s).

---

[1]LDL can be seen as learning from *soft* targets, but differs from soft labels in single-label classification: soft labels assume exactly one true class per instance, whereas LDL models an *instance-level* label distribution that may reflect *multiple* relevant labels. This instance-level distribution should not be confused with the *label-distribution* (class-frequency) heterogeneity across clients in federated learning.

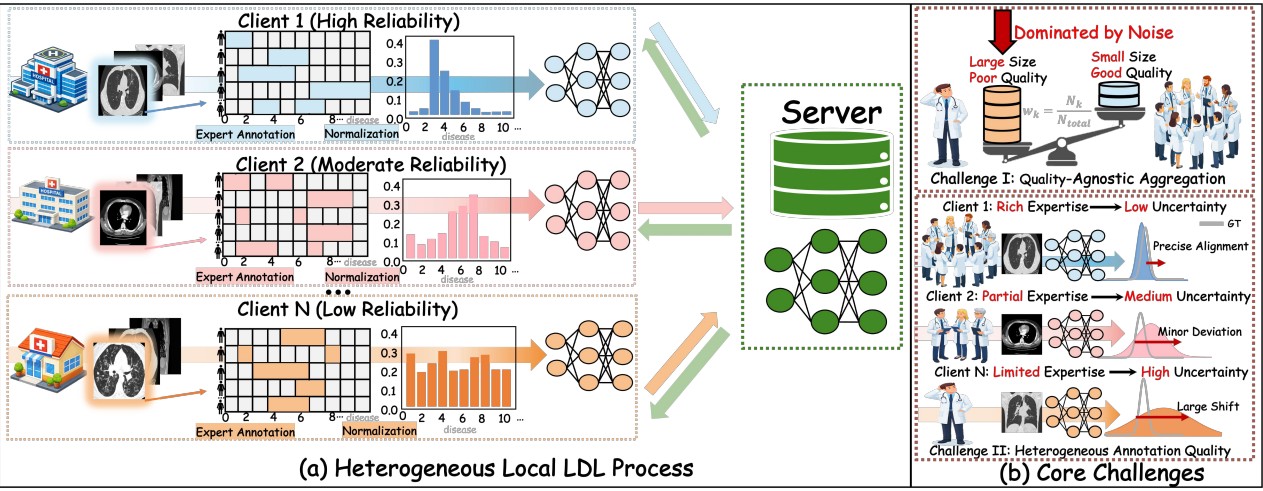

Figure 1. Illustration of the Federated Label Distribution Learning (Fed-LDL) framework and its inherent challenges. (a) The Fed-LDL Workflow: Clients with varying levels of reliability (e.g., top-tier hospitals vs. community clinics) locally optimize LDL models based on their private datasets, which exhibit diverse label distributions. (b) Core Challenges. Challenge I (Quality-Agnostic Aggregation): The server aggregates parameters based solely on sample size, allowing noisy clients with large data volumes to dominate the global model. Challenge II (Heterogeneous Annotation Quality): The intrinsic disparity in annotator expertise leads to inconsistent optimization targets, causing local models on low-quality clients to suffer from a "Large Shift" away from the precise ground truth.

depicted in Figure 1(b), the quality of data annotations varies significantly across clients, compounded by the challenge of inconsistent data distributions (non-IID). This raises a critical question: how can we leverage such AQH data to enhance local model performance while strictly adhering to the privacy constraint of not sharing raw data? Challenge ❏: Failure of Traditional Aggregation. As illustrated in the aggregation phase of Figure 1(b), conventional aggregation methods, such as FedAvg (McMahan et al., 2017), perform parameter weighting based on the sample sizes of different clients, implicitly assuming that each client is relatively reliable. However, this strategy becomes counterproductive when a client with a large volume of noisy data possesses significantly higher voting power than one with a small but high-quality dataset, leading to a substantial degradation in global model performance. Consequently, how to effectively balance the **quantity** and **quality** of client contributions remains a formidable challenge in Fed-LDL. Challenge ◆: Scarcity of Real-world Fed-LDL Benchmarks. To date, there is a lack of real-world Fed-LDL benchmarks for validating federated LDL methods under heterogeneous supervision quality, making rigorous empirical evaluation in realistic federated settings difficult.

In this paper, we propose *FedQual* to explicitly address the challenges arising from heterogeneous annotation quality. To mitigate the unreliability of local updates caused by uneven supervision quality, we incorporate a *global semantic anchor* as a guiding reference, applying stronger semantic correction to low-quality clients while preserving the autonomy of high-quality ones. To prevent unreliable updates from dominating the global model during aggregation,

FedQual proposes an *Effective Information Density* recalibration strategy that evaluates client contributions based on reliable information rather than raw sample size, enabling rational and quality-aware aggregation. Next, to bridge the lack of real-world Fed-LDL datasets, we introduce four new Fed-LDL benchmarks[2] (FER-LDL, FI-LDL, PIPAL-LDL, and KADID-LDL) and conduct extensive evaluations on them. Extensive experiments on these benchmarks demonstrate the effectiveness and robustness of the proposed framework. From a theoretical perspective, we prove that, under heterogeneous supervision quality, the optimal client-specific calibration strictly outperforms any optimal uniform calibration in Fed-LDL (i.e., the uniform strategy incurs a strictly larger total risk). This result provides a principled foundation for FedQual's quality-adaptive calibration and reliable-information-based aggregation. Our contributions are summarized as follows:

- To the best of our knowledge, this is the first work that systematically studies Fed-LDL: we identify its key challenges under heterogeneous supervision quality and propose a principled solution to resolve the resulting trust dilemma.

- We construct and release four Fed-LDL benchmarks (FER-LDL, FI-LDL, PIPAL-LDL, and KADID-LDL) with controlled annotation quality disparity, enabling rigorous and reproducible evaluation under heterogeneous supervision.

- From a theoretical perspective, we prove that under

---
[2]Benchmark construction details are provided in Section 4.

heterogeneous supervision quality, client-specific calibration is strictly better than any uniform calibration in Fed-LDL, providing a principled justification for our quality-adaptive design.

## 2. Related Work

### 2.1. Federated Learning under Data Heterogeneity

Federated learning enables collaborative training across decentralized clients while preserving privacy (Kairouz et al., 2021; Kou et al., 2026b). However, non-IID data poses major challenges, leading to gradient inconsistency and degraded global models (Liao et al., 2024; 2025b;a). To mitigate these issues, extensive research has explored two primary directions: optimization correction and representation alignment. Optimization-based methods such as Fed-Prox (Li et al., 2020) mitigate client drift by regularizing local updates. MOON (Li et al., 2021) leverages model-contrastive learning to align local feature representations, while FedGloSS (Caldarola et al., 2025) generates synthetic samples to compensate for visual distribution discrepancies.

Beyond statistical variations, the disparity in annotation quality introduces the challenge of heterogeneous label noise, where clients exhibit inconsistent noise levels. Fed-Div (Li et al., 2024) and FedFixer (Ji et al., 2024) identify and filter unreliable clients or samples under heterogeneous noise. FedENLC (Cho & Kim, 2026) introduces an end-to-end framework for noisy label correction, while FedES (Zeng et al., 2024) utilizes federated early-stopping to prevent models from memorizing noisy patterns. However, these approaches generally target discrete categorical noise. They typically treat noise as binary errors, making them ill-equipped for tasks requiring dense, continuous supervision, where noise manifests as subtle shifts in probability distributions rather than simple misclassifications.

### 2.2. Label Distribution Learning

Label Distribution Learning (LDL) has emerged as a paramount paradigm for quantifying label ambiguity. Unlike traditional classification frameworks that enforce definitive hard decisions, LDL models the label space as a probability distribution $\mathbf{d} \in \Delta^C$, where each element represents the description degree of a specific class to the instance (Geng, 2016). By capturing the relative importance of multiple labels, LDL supports fine-grained semantic interpretation in tasks involving subjective uncertainty, such as facial emotion recognition and medical image assessment (She et al., 2021; Le et al., 2023; Li et al., 2022; Chen et al., 2022; Zheng et al., 2024).

However, the efficacy of LDL hinges critically on the quality of the supervision signal. Reliable ground-truth distributions typically require aggregating diverse annotations from multiple annotators (Kou et al., 2023). To mitigate the impact of supervision noise, extensive research has focused on recovering reliable distributions from imperfect data. Existing approaches can be broadly categorized into structure-based and correlation-based methods. For instance, Manifold Learning techniques (Wang & Geng, 2023) exploit the local topological structure of the feature space to smooth out label noise, operating on the assumption that valid distributions should vary smoothly across the data manifold. Similarly, Label Correlation methods leverage global statistical dependencies among labels (e.g., the co-occurrence of distinct emotions) to rectify inconsistent predictions. Notable works (Ren et al., 2019) employ low-rank approximation to model these dependencies, while others (Xu & Zhou, 2017; Xu et al., 2025) utilize correlation decomposition mechanisms to effectively complete missing label information in sparse distribution settings.

Despite their effectiveness, these remediation strategies predominantly rely on a centralized data access assumption, where the algorithm can fully observe the entire dataset to compute global manifold structures or label correlation matrices. Therefore, standard Inaccurate LDL approaches are ill-suited for distributed environments where supervision quality varies but cross-client data inspection is prohibited.

### 2.3. Federated Label Distribution Learning

Federated Label Distribution Learning (Fed-LDL) focuses on training a global model to predict nuanced label distributions from decentralized data sources without sharing raw samples. Unlike conventional federated classification tasks, Fed-LDL introduces a unique challenge: the fidelity of the supervision signal varies substantially across clients due to *Annotation Quality Heterogeneity* (Kou et al., 2025c; 2026a). Local datasets are often annotated by users with varying levels of expertise, ranging from professional institutions to amateur users, which leads to systematically biased or noisy local supervision.

This heterogeneity creates a severe "Trust Dilemma" that breaks the assumptions of standard aggregation. Label distributions naturally exhibit dense, continuous semantics, where noise manifests as subtle shifts in probability mass rather than discrete label flips (Kou et al., 2025b). Consequently, conventional strategies that weight contributions strictly by sample size (e.g., FedAvg) become counterproductive, as they allow "large-but-noisy" clients to dominate the global model. Meanwhile, existing robust FL methods mainly target discrete categorical errors. On the other hand, centralized inaccurate-LDL methods rely on globally shared statistics that are unavailable in privacy-preserving settings.

Therefore, the core difficulty in Fed-LDL is to construct a trustworthy global model from distributed sources with disparate supervision quality. Effective Fed-LDL methods

must be capable of preventing noisy clients from degrading the aggregation, and ensuring that the global model is driven by high-fidelity knowledge.

## 3. Methodology

### 3.1. Notation

We consider a Fed-LDL system with $M$ clients $\mathcal{U} = \{u_1, \ldots, u_M\}$. Client $u_m$ holds $\mathcal{D}_m = \{(\mathbf{x}_{m,j}, \mathbf{d}_{m,j})\}_{j=1}^{N_m}$, where $\mathbf{d}_{m,j} \in \Delta^C$ is a label distribution ($\mathbf{d}_{m,j} \succeq \mathbf{0}$, $\|\mathbf{d}_{m,j}\|_1 = 1$). Let $f(\cdot; \mathbf{w}) : \mathcal{X} \to \Delta^C$ be the model. At communication round $t$, client $u_m$ maintains a local model $f(\cdot; \mathbf{w}_m^t)$ with parameters $\mathbf{w}_m^t$; given an input $\mathbf{x}$, it outputs logits $\mathbf{z}_m^t(\mathbf{x}) \in \mathbb{R}^C$ and the predicted label distribution is $\mathbf{p}_m^t(\mathbf{x}) = \mathrm{softmax}(\mathbf{z}_m^t(\mathbf{x}))$. We assign a *Quality Indicator* $q_m \in \mathbb{R}^+$ to each client $u_m$, $q_m$ quantifies the statistical fidelity of the local dataset $\mathcal{D}_m$.

> **Overview.** To address the two challenges in Section 1, we propose FedQual, which mitigates the trust dilemma in Fed-LDL from both the client and server sides: (i) client-side trust rectification via quality-adaptive client training under the GSA to avoid unreliable update directions; and (ii) server-side trust filtering via progressively reweighting clients by effective reliable information to prevent "large-but-noisy" clients from dominating aggregation.

### 3.2. Can We Trust Local client Updates in Fed-LDL?

As discussed in Section 1, Fed-LDL faces a trust dilemma. Beyond Non-IID data, clients exhibit annotation quality heterogeneity, so local gradients from low-quality clients may be unreliable. Using the server model for correction is also risky: when the server distribution differs greatly from a client's data, the "corrected" gradient can be biased. Hence, a core problem in Fed-LDL is to find a trustworthy update direction for each client.

To address this issue, we introduce a new concept in the local-client training stage, namely the Global Semantic Anchor (GSA), which is used to appropriately calibrate different clients. The rationale for this design lies in the *ensemble consensus theory* (Lin et al., 2020; Li & Wang, 2019): analogous to Bayesian model ensembling (Chen & Chao, 2021), aggregating diverse local models cancels out idiosyncratic annotation noise while reinforcing shared semantic patterns. Functioning as a robust supervisor grounded in this consensus, the GSA intervenes only when a local client deviates from the generalized update direction. We next define the

GSA as follows.

$$\mathcal{A}(\mathbf{x}) \triangleq \mathbf{z}(\mathbf{x}; \mathbf{w}_g^t), \quad \text{with } \mathbf{w}_g^t = \mathrm{Agg}(\mathbf{w}_1^{t-1}, \ldots, \mathbf{w}_M^{t-1}). \tag{1}$$

where $\mathcal{A}(\cdot) \in \mathbb{R}^C$ denotes the Global Semantic Anchor in *logits space*, instantiated as the global-model logits $\mathbf{z}(\mathbf{x}; \mathbf{w}_g^t)$ on input $\mathbf{x}$. Here, $\mathbf{w}_g^t$ is the global model parameter at round $t$, obtained by aggregating the client models from the previous round, i.e., $\mathbf{w}_g^t = \mathrm{Agg}(\mathbf{w}_1^{t-1}, \ldots, \mathbf{w}_M^{t-1})$, and then broadcast to all clients at the start of round $t$.

### 3.3. Take the Strengths, Compensate the Weaknesses

In Fed-LDL, clients have different supervision accuracy levels, denoted by $q_m$. Our principle is simple: *let accurate clients be themselves, and help inaccurate clients more*. Specifically, when $q_m$ is large, the local signal is trustworthy and the GSA should intervene less to preserve client autonomy; when $q_m$ is small, the local signal is unreliable and the GSA should step in more aggressively to calibrate the update direction by aligning the client output with the anchor. To express *different degrees of alignment* in a more principled way, we formulate each client's local training as a learning and calibration joint objective:

$$\min_{\mathbf{w}_m} \mathbb{E}_{\mathbf{x} \sim \mathcal{D}_m} \Big[ (1 - \alpha_m) \underbrace{\ell(\mathbf{d}_m(\mathbf{x}), f(\mathbf{x}; \mathbf{w}_m))}_{\text{local learning}}$$
$$+ \alpha_m \underbrace{\mathcal{R}(\mathcal{A}(\mathbf{x}), \mathbf{z}_m(\mathbf{x}; \mathbf{w}_m))}_{\text{anchor calibration}} \Big], \tag{2}$$

where $\mathbf{w}_m$ denotes the parameters of client $m$, and $\mathcal{D}_m$ is its local dataset (with the expectation taken over the empirical distribution). $\mathbf{d}_m(\mathbf{x}) = [d_{m,1}(\mathbf{x}), \ldots, d_{m,C}(\mathbf{x})]^\top$ is the label-distribution supervision on client $m$, and $f(\mathbf{x}; \mathbf{w}_m)$ denotes the client prediction (a probability distribution after softmax). The first term (*local learning*) fits the model to the local label distribution via the LDL loss $\ell(\cdot, \cdot)$ (e.g., $\mathrm{KL}(\mathbf{d}_m(\mathbf{x}) \| f(\mathbf{x}; \mathbf{w}_m))$). The second term (*anchor calibration*) uses the Global Semantic Anchor $\mathcal{A}(\mathbf{x})$ to rectify unreliable local updates by aligning the client logits $\mathbf{z}_m(\mathbf{x}; \mathbf{w}_m)$ with the anchor in logits space through $\mathcal{R}(\cdot, \cdot)$. And $\alpha_m$ is defined as

$$\begin{cases} \alpha_m \triangleq \sigma\Big(\beta\big(\lambda(q_m) - \lambda_0\big)\Big), \\ \lambda(q_m) = \max\Big(0, 1 - \dfrac{q_m}{\tau}\Big). \end{cases} \tag{3}$$

where $\sigma(\cdot)$ is the sigmoid function, $\beta > 0$ controls the sharpness of the transition, and $\lambda_0$ is an intervention offset. Since $\lambda(q_m)$ decreases with $q_m$, $\alpha_m$ becomes larger for low-quality clients (small $q_m$), yielding stronger anchor calibration; meanwhile, $\alpha_m$ stays small for high-quality clients, so the objective is dominated by local learning. In this way, the $m$-th client is trained by *learning + correction* jointly, with the correction strength automatically adapted to its supervision accuracy.

## 3.4. Trust First, Then Scale: Quality-to-Quantity Annealed Aggregation

This trust issue also breaks standard aggregation: FedAvg-style sample-size weighting is unreasonable when large clients have low-quality annotations, allowing unreliable updates to dominate the global model. This is *stage-dependent*: early on, the anchor and calibration are still unstable, so the server should emphasize *quality-aware* reweighting to suppress "large-but-noisy" clients; later, as the anchor stabilizes and local updates become more trustworthy, aggregation can gradually shift toward *quantity-aware* weighting for statistical efficiency. Motivated by this, we propose a progressive aggregation strategy that anneals from quality-dominated to quantity-dominated weighting. We first define the *effective reliable information* of client $m$ as

$$S_m \triangleq N_m \cdot q_m, \tag{4}$$

where $S_m$ denotes the effective reliable information provided by client $m$. To enable a smooth transition over rounds, we introduce a round-dependent *trust annealing* factor $\rho_t \in [0, 1]$ and define an intermediate score

$$\widetilde{S}_m^t \triangleq \left(S_m\right)^{1-\rho_t} \left(N_m\right)^{\rho_t} = N_m \cdot q_m^{1-\rho_t}, \tag{5}$$

where $\rho_t$ controls the transition from quality-aware to quantity-aware weighting, and $\widetilde{S}_m^t$ is the round-$t$ score used for aggregation. The aggregation weight is then obtained by a normalized soft weighting:

$$\omega_m^t \triangleq \frac{\exp\left(\gamma_{temp} \log \widetilde{S}_m^t\right)}{\sum_{j=1}^M \exp\left(\gamma_{temp} \log \widetilde{S}_j^t\right)} = \frac{\left(\widetilde{S}_m^t\right)^{\gamma_{temp}}}{\sum_{j=1}^M \left(\widetilde{S}_j^t\right)^{\gamma_{temp}}}, \tag{6}$$

where $\omega_m^t$ is the aggregation weight assigned to client $m$ at round $t$, and $\gamma_{temp} > 0$ controls the sharpness (inverse-temperature) of the weighting. Finally, the server updates the global model by

$$\mathbf{w}_g^{t+1} = \sum_{m=1}^M \omega_m^t \, \mathbf{w}_m^{t+1}, \tag{7}$$

where $\mathbf{w}_g^{t+1}$ denotes the aggregated global model for the next round. When $\rho_t = 0$, we have $\widetilde{S}_m^t = S_m = N_m q_m$, yielding a fully quality-aware aggregation that strongly down-weights low-quality clients even if they have large $N_m$. When $\rho_t \to 1$, $\widetilde{S}_m^t \to N_m$, so the rule smoothly recovers sample-size weighting, which becomes appropriate once anchor-based calibration stabilizes and local updates are more trustworthy. In practice, $\rho_t$ can be implemented as an increasing schedule (e.g., linear warm-up) to reflect the progressively improved reliability of the anchor.

## 4. Theoretical Analysis

Recall that our local update combines *local learning* and *anchor calibration* with a client-specific calibration weight.[3] This design directly matches the trust dilemma in Fed-LDL: clients have heterogeneous supervision quality, so the reliability of their local updates varies substantially. A natural question is whether we can instead use a *uniform* calibration strength for all clients. We show that, under such heterogeneity, treating all clients equally is provably suboptimal, thereby justifying the effectiveness of our quality-adaptive modulation used in FedQual.

**Theorem 4.1** (Adaptive calibration is strictly better than uniform calibration). *For client $m$, consider the expected excess risk $\mathcal{R}_m(\lambda)$ of applying anchor calibration with strength $\lambda \in [0, 1]$ under the local quadratic surrogate in Eq. (A.11). Define $\mathcal{J}_{adapt} \triangleq \sum_{m=1}^M \min_{\lambda_m \in [0,1]} \mathcal{R}_m(\lambda_m)$ and $\mathcal{J}_{uni} \triangleq \min_{\bar{\lambda} \in [0,1]} \sum_{m=1}^M \mathcal{R}_m(\bar{\lambda})$. If the optimal trade-offs are not identical across clients, i.e., there exist $m \neq n$ such that $\lambda_m^\star \neq \lambda_n^\star$ (with $\lambda_m^\star$ given in Eq. (A.12)), then the adaptive strategy achieves strictly smaller total risk:*

$$\mathcal{J}_{adapt} < \mathcal{J}_{uni}. \tag{8}$$

Theorem 4.1 proves that a *uniform* calibration strength is strictly suboptimal under client heterogeneity, whereas *client-specific* calibration achieves lower total risk. Practically, it prevents over-correcting high-quality clients while providing stronger rectification for low-quality ones, leading to a more stable and accurate federation. The detailed proof of Theorem 4.1 is in Appendix A.

**Remark.** Our method performs anchor calibration via a logits-level regularizer rather than explicitly interpolating parameters. For theoretical tractability, we adopt a standard surrogate view that this calibration *implicitly pulls* the client solution toward the global anchor, which motivates the interpolation model in Eq. (A.10).

## 5. Benchmark Construction

Existing LDL datasets typically assume uniformly reliable supervision, thus failing to reflect the *annotation quality disparity* in federated scenarios; consequently, Fed-LDL lacks rigorous benchmarks to stress-test methods under heterogeneous supervision. To fill this gap, we establish *four new Fed-LDL benchmarks* covering emotion recognition (FER-LDL, FI-LDL) and IQA (PIPAL-LDL, KADID-LDL) by *re-annotating* existing datasets with dense label distributions (Goodfellow et al., 2015; You et al., 2016; Gu et al., 2020; Lin et al., 2019). We adopt a rigorous *10-expert*

---

[3] In Sec. 2, this weight is denoted by $\alpha_m$; in this section we use $\lambda$ to denote the same intervention strength for analysis convenience, i.e., $\lambda \equiv \alpha$.

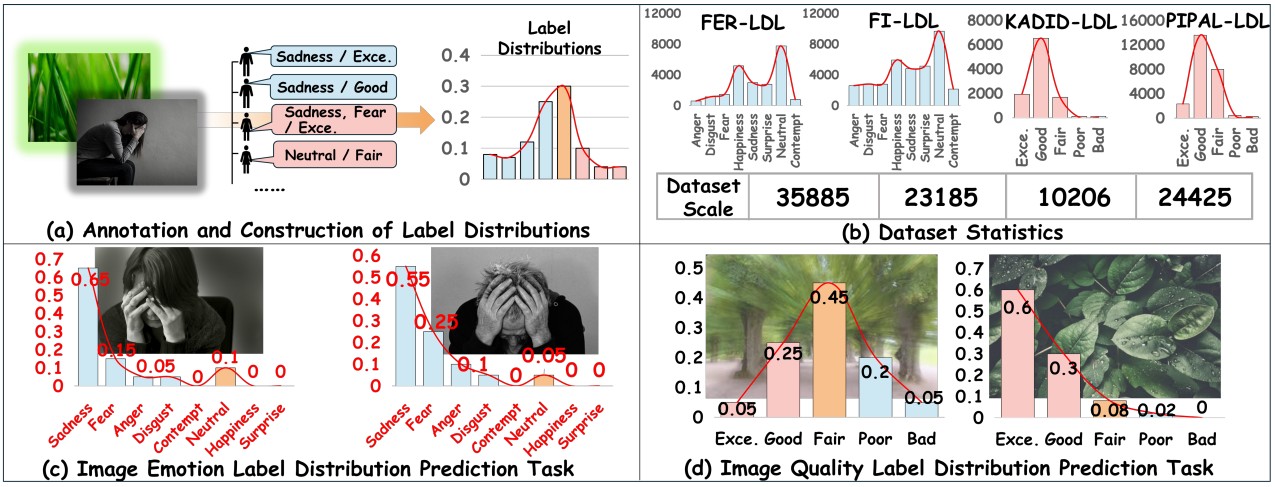

*Figure 2.* **Overview of our Fed-LDL benchmarks.** (a) 10-expert ensemble annotation and label-distribution construction, yielding the latent ground-truth distribution $\mathbf{d}_{GT}$. (b) Dataset statistics, including dataset scale and aggregated label-distribution profiles for FER-LDL, FI-LDL, KADID-LDL, and PIPAL-LDL. (c) A qualitative emotion-recognition example with an 8-class distribution, illustrating cognitive ambiguity. (d) A qualitative IQA example with a 5-bin score distribution, illustrating inherent subjectivity.

*human-ensemble* protocol to build latent ground-truth distributions $\mathbf{d}_{GT}$ by aggregating and normalizing expert votes (details in the appendix). For emotion recognition, experts provide multi-hot judgments over 8 emotions and votes are aggregated into $\mathbf{d}_{GT}$; for IQA, discrete ratings are converted to vote vectors and aggregated into a 5-bin score distribution. These benchmarks provide a practical testbed for evaluating Fed-LDL under the trust dilemma.

*Task description.* As illustrated in Figure 2(c–d), our benchmarks cover two label-distribution prediction tasks: (i) *image quality assessment (IQA)*, where the goal is to predict a 5-bin perceptual quality distribution over {Excellent, Good, Fair, Poor, Bad}; and (ii) *facial emotion recognition*, where the goal is to predict an 8-class emotion distribution capturing the mixture of plausible emotions in the image.

*Dataset Statistics and Annotation Details.* We first report the overall statistics of our four Fed-LDL benchmarks, including the dataset scale and the aggregated label-distribution profiles. Specifically, Figure 2(b) summarizes the number of samples in each benchmark and visualizes the *summed* label distributions (i.e., cumulative label mass over all instances). Notably, the total mass across different labels is highly imbalanced, exhibiting substantial heterogeneity among labels and across datasets, which further increases the difficulty of robust Fed-LDL under heterogeneous supervision. We then describe the annotation procedure used to construct high-fidelity latent ground-truth distributions. As shown in Figure 2(a), each image is independently annotated by 10 human experts, and their votes are aggregated and normalized to form $\mathbf{d}_{GT}$ [4].

---

[4]detailed annotation guidelines and the expert protocol are deferred to the appendix.

## 6. Experiment

### 6.1. Setups

**Baseline Methods.** We compare *FedQual* against three categories of baselines. **(i) Optimization-Stabilization:** *FedAvg* (McMahan et al., 2017) performs sample-size weighted averaging and assumes uniform client reliability. *FedProx* (Li et al., 2020) improves it with a proximal term to restrict local model drift. *(ii) Representation-Alignment:* *MOON* (Li et al., 2021) employs contrastive loss to align local and global representations. *FedRDN* (Yan et al., 2025) uses robust distance metrics to filter unreliable updates. *FedGloSS* (Caldarola et al., 2025) generates synthetic samples to mitigate feature skew. *(iii) Quality-Aware:* To disentangle the contribution of $q_m$, we introduce *FedQAgg*, which applies quality-weighted aggregation at the server, and *FedQRect*, which introduces client-side regularization proportional to data quality. Both serve as component-level baselines to validate the design of FedQual.

**Evaluation Metrics.** Following the standard protocol in LDL, we employ six diverse metrics to comprehensively evaluate the alignment between the predicted and ground-truth distributions. These metrics are categorized into two groups: (1)Distance Measures ($\downarrow$): *Chebyshev*, *Clark*, *Canberra*, and *Kullback-Leibler divergence*, where lower values indicate better performance; and (2)Similarity Measures ($\uparrow$): *Cosine coefficient* and *Intersection similarity*, where higher values imply greater consistency (Geng, 2016).

**Non-IID and Noise Simulation Settings.** To simulate realistic federated environments, we construct data heterogeneity along two axes: label distribution skew and annotation quality variation. *Non-IID Partitioning.* We adopt a two-

stage strategy to induce label distribution non-IIDness. First, we discretize the soft ground-truth $d_{GT}$ into sparse multi-label distributions using a Top-$K$ operation, retaining only the top $K$ labels per sample ($K \in \{1, 2, 3, 4\}$). Then, client partitions are created based on a Dirichlet distribution with concentration parameter $\gamma \in \{0.25, 0.5, 0.75, 1.0\}$ to ensure heterogeneous label support across clients. *Annotation Noise Injection.* We simulate annotation quality heterogeneity in two ways. (1) To degrade local label fidelity, we vary the number of annotators used to generate each label distribution; fewer annotators yield higher uncertainty and thus lower quality scores $q_m$. (2) To introduce global noise imbalance, we control the proportion of low-quality clients $\rho_{noise} \in \{0.25, 0.5, 0.75\}$ by assigning noisy supervision patterns to a subset of clients. This dual noise modeling reflects both intra-client and inter-client annotation variability.

## 6.2. Results and Findings

**Comparative Analysis with State-of-the-Art.** The comparative results are summarized in Table 1. From the table, we can draw the following conclusions:

- FedQual delivers the best overall performance across four benchmarks under six metrics, consistently outperforming both quality-agnostic FL baselines (e.g., FedAvg/FedProx/MOON) and quality-aware competitors (e.g., FedGloSS/FedQAgg/FedQRect).

- FedQual achieves the lowest KL divergence on FER-LDL and FI-LDL (0.2528 and 0.4085, respectively) and the highest Cosine similarity, indicating strong robustness to heterogeneous annotation noise.

- FedQual maintains clear advantages on IQA, reaching a KL as low as 0.0908 on KADID-LDL and achieving Cosine similarity above 0.96, demonstrating reliable distribution learning under intrinsic subjectivity.

- The improvements hold for both distance-based criteria (KL/Chebyshev/Clark/Canberra) and similarity-based criteria (Intersect/Cosine), suggesting that FedQual enhances not only pointwise accuracy but also the overall shape alignment of predicted label distributions.

- Compared with methods that focus on only one side (either local robustness or aggregation), FedQual's *client-side quality-adaptive rectification* and *server-side reliability-aware aggregation* work synergistically, which is crucial for resolving the Fed-LDL trust dilemma.

**Ablation Study.** We conduct ablations on FER-LDL and PIPAL-LDL (Table 2) to evaluate the individual contribution of each component within the FedQual framework. Starting from FedAvg: **+A** adds the client-side rectifier, and **+A+B** further adds server-side reweighting. We observe:

- **+A** often improves over FedAvg on both datasets, confirm-

ing the benefit of anchor-guided local calibration.

- **+A+B** further improves over **+A** and achieves the best overall performance, showing the complementarity between rectification and reliability-aware aggregation.

- On PIPAL-LDL, the full model reduces KL from 0.252 to 0.116 and increases Cosine from 0.929 to 0.958.

**Robustness to Annotation Quality Heterogeneity.** We evaluate the robustness of FedQual against two dimensions of annotation quality degradation: reduced label fidelity and increased prevalence of noisy clients. As shown in Figure 3, we first vary the quality indicator $q_m$ to simulate scenarios with fewer annotators per client. Despite this degradation in supervision, FedQual maintains consistently low KL divergence and high Intersection scores across all four benchmarks, demonstrating strong resilience to noise intensity. We further examine the effect of increasing the ratio of low-quality clients ($\rho_{noise} \in \{0.25, 0.5, 0.75\}$). Results show that even when 75% of clients are noisy, the performance of FedQual remains remarkably stable across all evaluation metrics. These findings confirm that FedQual is robust to both localized label noise and global client imbalance.

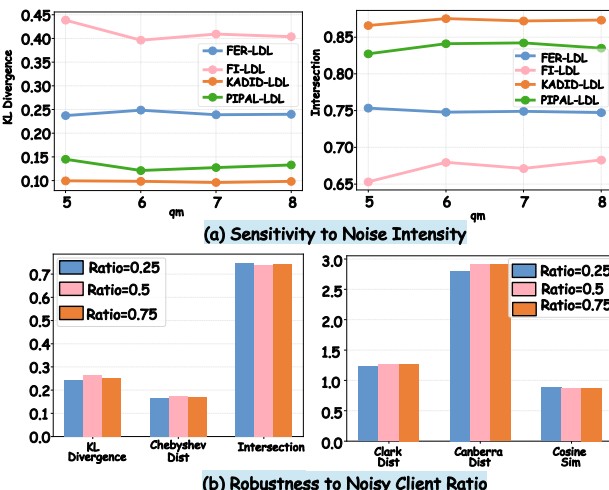

*Figure 3.* Robustness analysis against Annotation Quality Heterogeneity. **(a) Sensitivity to Noise Intensity:** The impact of varying the quality indicator $q_m \in \{5, 6, 7, 8\}$ on the four benchmarks. The flat curves indicate that FedQual maintains high performance regardless of the local noise variance. **(b) Impact of Noisy Client Ratio:** Performance comparison under different ratios of low-quality clients ($\rho_{noise} \in \{0.25, 0.50, 0.75\}$). FedQual exhibits strong resistance to noise domination, sustaining stable metrics even when 75% of clients are noisy.

**Label Distribution Skew Robustness.** We assess the robustness of FedQual under varying levels of label distribution skew, controlled by the Dirichlet parameter $\gamma \in \{0.25, 0.5, 0.75, 1.0\}$ on the FER-LDL benchmark (Figure 4). Lower $\gamma$ values induce more severe inter-client label imbalance. Despite this, FedQual exhibits stable perfor-

*Table 1.* Performance comparison on four LDL benchmarks. ↓ indicates lower is better, while ↑ indicates higher is better. The best results are highlighted in blue and the second-best results in green. Our method, FedQual, consistently achieves competitive performance.

| Dataset | Method | KL ↓ | Chebyshev ↓ | Clark ↓ | Canberra ↓ | Intersect ↑ | Cosine ↑ |
|---|---|---|---|---|---|---|---|
| **FER-LDL** | FedAvg | 0.3420 | 0.2134 | 1.3230 | 3.0968 | 0.6961 | 0.8162 |
| | FedProx | 0.3007 | 0.1956 | 1.2742 | 2.9578 | 0.7135 | 0.8414 |
| | FedRDN | 0.3593 | 0.2161 | 1.3565 | 3.2050 | 0.6840 | 0.8057 |
| | MOON | 0.3782 | 0.2329 | 1.3294 | 3.1303 | 0.6745 | 0.7947 |
| | FedGloSS | 0.3155 | 0.1982 | 1.3647 | 3.2116 | 0.7137 | 0.8414 |
| | FedQAgg | 0.2756 | 0.1775 | 1.2942 | 2.9925 | 0.7329 | 0.8603 |
| | FedQRect | 0.2638 | 0.174 | 1.2812 | 2.9513 | 0.7385 | 0.8678 |
| | **FedQual** | **0.2528** | **0.1716** | **1.2603** | **2.8874** | **0.7447** | **0.8739** |
| **FI-LDL** | FedAvg | 0.4233 | 0.2283 | 1.8579 | 4.5126 | 0.6673 | 0.8102 |
| | FedProx | 0.5119 | 0.2573 | 1.8655 | 4.5795 | 0.6194 | 0.7873 |
| | FedRDN | 0.5434 | 0.2733 | 1.8560 | 4.5758 | 0.5919 | 0.7660 |
| | MOON | 0.4493 | 0.2306 | 1.8574 | 4.5259 | 0.6516 | 0.8025 |
| | FedGloSS | 0.4843 | 0.2414 | 1.8599 | 4.5235 | 0.6385 | 0.7939 |
| | FedQAgg | 0.5451 | 0.2514 | 1.8769 | 4.5897 | 0.6368 | 0.7922 |
| | FedQRect | 0.482 | 0.2352 | 1.8688 | 4.5429 | 0.6533 | 0.8022 |
| | **FedQual** | **0.4085** | **0.2167** | **1.8545** | **4.4802** | **0.6785** | **0.8173** |
| **KADID-LDL** | FedAvg | 0.1248 | 0.1435 | 1.1949 | 2.7169 | 0.8532 | 0.9547 |
| | FedProx | 0.1090 | 0.1391 | 1.2277 | 2.6052 | 0.8560 | 0.9600 |
| | FedRDN | 0.1176 | 0.1393 | 1.3334 | 2.6642 | 0.8521 | 0.9604 |
| | MOON | 0.4262 | 0.1492 | 1.5034 | 2.6538 | 0.8414 | 0.9446 |
| | FedGloSS | 0.1301 | 0.1445 | 1.4998 | 2.6884 | 0.8501 | 0.9537 |
| | FedQAgg | 0.1140 | 0.1316 | **0.9655** | 2.6231 | 0.8656 | 0.9607 |
| | FedQRect | 0.1155 | 0.1445 | 0.9924 | 2.6413 | 0.8522 | 0.9559 |
| | **FedQual** | **0.0908** | **0.1201** | 1.0682 | **2.5718** | **0.8771** | **0.9672** |
| **PIPAL-LDL** | FedAvg | 0.2521 | 0.1676 | 1.3976 | 2.5095 | 0.8156 | 0.9292 |
| | FedProx | 0.1203 | 0.1410 | 1.3990 | 2.4085 | 0.8436 | 0.9519 |
| | FedRDN | 0.2186 | 0.1966 | 1.4334 | 2.6590 | 0.7680 | 0.9063 |
| | MOON | 0.1282 | 0.1457 | 1.4151 | 2.4547 | 0.8406 | 0.9469 |
| | FedGloSS | 0.1393 | 0.1538 | 1.4077 | 2.4273 | 0.8322 | 0.9410 |
| | FedQAgg | 0.4398 | 0.1660 | 1.3958 | 2.4859 | 0.8151 | 0.9287 |
| | FedQRect | 0.1543 | 0.1659 | 1.3993 | 2.4814 | 0.8142 | 0.937 |
| | **FedQual** | **0.1162** | **0.1389** | **1.3906** | **2.3868** | **0.8442** | **0.9580** |

mance across all metrics. Specifically, metrics such as KL divergence and Cosine similarity remain nearly unchanged as $\gamma$ varies, indicating that FedQual is resilient to non-IID label distributions. This confirms its applicability in real-world settings where client data is often label-skewed.

**Robustness to Label Multiplicity.** To simulate challenging label heterogeneity across clients, we first degrade the original soft label distribution $d_{GT}$ into a sparse multi-label form using a Top-$K$ operation, where $K \in \{1, 2, 3, 4\}$ determines the number of retained labels per sample. Based on these multi-label annotations, we partition data across clients such that each client supports a distinct subset of labels, thereby introducing non-IID feature distributions. As

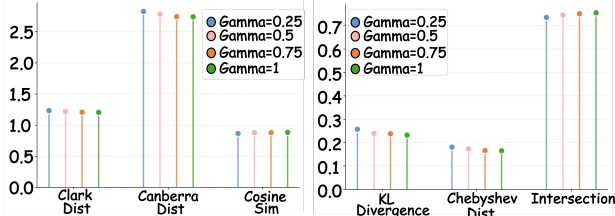

*Figure 4.* Sensitivity analysis of the label distribution skew parameter $\gamma$ on the FER-LDL benchmark. We evaluate the performance across varying skew levels $\gamma \in \{0.25, 0.5, 0.75, 1.0\}$. The results across six metrics demonstrate that our proposed FedQual framework remains robust to different degrees of label distribution skew.

*Table 2.* Ablation study on FER-LDL and PIPAL-LDL datasets. "Base" denotes the standard FedAvg. "+A" incorporates the *Quality-Modulated Rectifier* to mitigate local drift. "+A+B" denotes the full framework. The background colors distinguish different module configurations. **Bold** indicates the best performance.

| Metric | FER-LDL | | | PIPAL-LDL | | |
|---|---|---|---|---|---|---|
| | Base | +A | +A+B | Base | +A | +A+B |
| KL ↓ | 0.342 | 0.264 | **0.253** | 0.252 | 0.154 | **0.116** |
| Cheb ↓ | 0.213 | 0.174 | **0.172** | 0.168 | 0.166 | **0.139** |
| Clark ↓ | 1.323 | 1.281 | **1.260** | 1.398 | 1.399 | **1.391** |
| Canb ↓ | 3.097 | 2.951 | **2.887** | 2.510 | 2.481 | **2.387** |
| Inter ↑ | 0.696 | 0.739 | **0.745** | 0.816 | 0.814 | **0.844** |
| Cos ↑ | 0.816 | 0.868 | **0.874** | 0.929 | 0.937 | **0.958** |

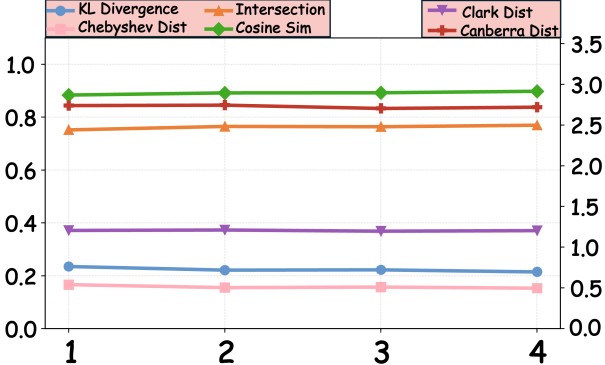

*Figure 5.* Impact of partition label multiplicity (Top-$K$) on FER-LDL. We vary the Top-$K$ parameter ($K \in \{1, 2, 3, 4\}$) used to discretize $\mathbf{d}_{GT}$ for simulating Non-IID feature skew. The flat trends demonstrate that FedQual maintains robust performance regardless of the label density used for client partitioning.

shown in Figure 5, FedQual maintains consistently stable performance across all six evaluation metrics, regardless of the label density. These results demonstrate that FedQual is robust to variations in label multiplicity and effectively adapts to diverse label configurations across the federation.

**Scalability to Federation Size.** We evaluate the scalability of FedQual by varying the total number of participating clients in the federation from 25 to 100 on the FI-LDL benchmark. As shown in Figure 6, the performance remains highly stable across all six evaluation metrics, including KL divergence, Intersection, and Cosine similarity. This indicates that FedQual can seamlessly adapt to larger federations without introducing performance degradation. The results highlight its practical applicability to real-world federated systems with a growing number of participants.

**Robustness to Partial Participation.** We evaluate the robustness of FedQual under varying client participation ra-

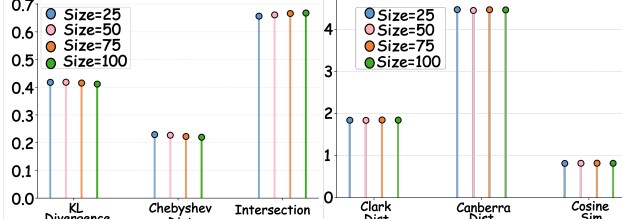

*Figure 6.* Scalability analysis regarding the federation size on the FI-LDL benchmark. We evaluate the model performance by varying the total client pool size in the range of $\{25, 50, 75, 100\}$. The consistent results demonstrate that FedQual scales effectively to larger federations without performance degradation.

tios to simulate realistic deployment scenarios with intermittent or limited client availability. Specifically, we vary the active client ratio per communication round $\rho_{online} \in \{0.2, 0.4, 0.6, 0.8\}$ on the KADID-LDL benchmark. As shown in Figure 7, FedQual achieves consistently stable performance across all six evaluation metrics, even when only 20% of clients participate per round. The flat performance trends confirm that our framework is robust to partial participation and can maintain effectiveness in practical federated environments with unreliable or sparse client engagement.

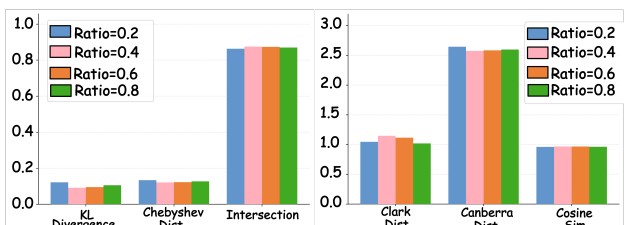

*Figure 7.* Robustness analysis regarding the client participation ratio on KADID-LDL. We evaluate model performance by varying the ratio of active clients per round in the range of $\{0.2, 0.4, 0.6, 0.8\}$. The consistent performance across all six metrics demonstrates that our framework remains effective even under low participation rates.

## 7. Conclusion

In this work, we tackle the critical but under-explored challenge of *Annotation Quality Heterogeneity* in Fed-LDL. We propose FedQual, a synergistic framework that decouples robust optimization: locally, it employs a *Quality-Modulated Rectifier* leveraged by a *Global Semantic Anchor* to correct optimization drift; globally, it utilizes a *Quality-Weighted Filter* to ensure high-fidelity knowledge drives the aggregation. To facilitate research, we also release four benchmarks (FER-LDL, FI-LDL, PIPAL-LDL, KADID-LDL) featuring a rigorous Dual Heterogeneity simulation protocol that captures both label distribution skew and varied annotation disparity. Extensive experiments confirm that FedQual establishes a new state-of-the-art.

# Acknowledgements

This research was supported by the Jiangsu Science Foundation (BG2024036, BK20243012), the National Science Foundation of China (62125602, U24A20324, 92464301), the New Cornerstone Science Foundation through the XPLORER PRIZE, and the Fundamental Research Funds for the Central Universities (2242025K30024 supported by the National). This work was Natural Science Foundation of China under Grant U24A20322 and Grant 62576094. This research work is also supported by the Big Data Computing Center of Southeast University.

# Impact Statement

This work advances privacy-preserving semantic learning by introducing the FedQual framework and four new Fed-LDL benchmarks , which may benefit sensitive applications such as medical image assessment and mental-health analysis, but could also be misused for unauthorized surveillance or emotional profiling if applied without appropriate ethical safeguards.

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

# Appendix Table of Contents

# A. Theoretical Analysis

Recall that our local update combines *local learning* and *anchor calibration* with a client-specific calibration weight.[5] This design directly matches the trust dilemma in Fed-LDL: clients have heterogeneous supervision quality, so the reliability of their local updates varies substantially. A natural question is whether we can instead use a *uniform* calibration strength for all clients. We show that, under such heterogeneity, treating all clients equally is provably suboptimal, thereby justifying the effectiveness of our quality-adaptive modulation used in FedQual.

**Theorem A.1** (Adaptive calibration is strictly better than uniform calibration). *For client $m$, consider the expected excess risk $\mathcal{R}_m(\lambda)$ of applying anchor calibration with strength $\lambda \in [0, 1]$ under the local quadratic surrogate in Eq. (A.11). Define $\mathcal{J}_{adapt} \triangleq \sum_{m=1}^{M} \min_{\lambda_m \in [0,1]} \mathcal{R}_m(\lambda_m)$ and $\mathcal{J}_{uni} \triangleq \min_{\bar{\lambda} \in [0,1]} \sum_{m=1}^{M} \mathcal{R}_m(\bar{\lambda})$. If the optimal trade-offs are not identical across clients, i.e., there exist $m \neq n$ such that $\lambda_m^\star \neq \lambda_n^\star$ (with $\lambda_m^\star$ given in Eq. (A.12)), then the adaptive strategy achieves strictly smaller total risk:*

$$\mathcal{J}_{adapt} < \mathcal{J}_{uni}. \tag{A.9}$$

Theorem A.1 proves that a *uniform* calibration strength is strictly suboptimal under client heterogeneity, whereas *client-specific* calibration achieves lower total risk. Practically, it prevents over-correcting high-quality clients while providing stronger rectification for low-quality ones, leading to a more stable and accurate federation.

*Proof.* For tractability, we model the rectified outcome of client $m$ by a convex interpolation in parameter space,

$$\mathbf{w}_m^{rect}(\lambda) \triangleq (1 - \lambda)\mathbf{w}_m^{loc} + \lambda\,\mathbf{w}_g^t, \tag{A.10}$$

where $\mathbf{w}_m^{loc}$ denotes the (hypothetical) local solution induced by clean supervision on client $m$, and $\mathbf{w}_g^t$ is the global model parameter (which induces the anchor) broadcast at round $t$. Assuming the loss is locally smooth and strongly convex around the client-specific optimum $\mathbf{w}_m^\star$, the expected excess risk of $\mathbf{w}_m^{rect}(\lambda)$ admits the following bias–variance surrogate:

$$\mathcal{R}_m(\lambda) \approx (1 - \lambda)^2\,\sigma_m^2 + \lambda^2\,\delta_m^2 + C, \tag{A.11}$$

where $\sigma_m^2 \triangleq \mathbb{E}\big[\|\mathbf{w}_m^{loc} - \mathbf{w}_m^\star\|_2^2\big]$ captures the variance induced by local supervision noise, $\delta_m^2 \triangleq \|\mathbf{w}_g^t - \mathbf{w}_m^\star\|_2^2$ captures the bias caused by semantic shift between the anchor and the client-specific optimum, and $C$ is independent of $\lambda$. Since $\mathcal{R}_m(\lambda)$ is a strictly convex quadratic in $\lambda$, its unique minimizer is

$$\lambda_m^\star = \arg\min_{\lambda \in [0,1]} \mathcal{R}_m(\lambda) = \frac{\sigma_m^2}{\sigma_m^2 + \delta_m^2}, \tag{A.12}$$

(clipped to $[0, 1]$ if necessary). Moreover, $\mathcal{R}_m(\lambda)$ can be written in completed-square form as

$$\mathcal{R}_m(\lambda) = \mathcal{R}_m(\lambda_m^\star) + (\sigma_m^2 + \delta_m^2)\,(\lambda - \lambda_m^\star)^2, \tag{A.13}$$

up to a $\lambda$-independent constant. Therefore, for any uniform $\bar{\lambda}$,

$$\sum_{m=1}^{M} \mathcal{R}_m(\bar{\lambda}) = \underbrace{\sum_{m=1}^{M} \mathcal{R}_m(\lambda_m^\star)}_{\mathcal{J}_{adapt}} + \underbrace{\sum_{m=1}^{M} (\sigma_m^2 + \delta_m^2)\,(\bar{\lambda} - \lambda_m^\star)^2}_{\text{excess risk}}. \tag{A.14}$$

The excess-risk term is a sum of nonnegative quadratic terms and is strictly positive when $\exists\, m \neq n$ such that $\lambda_m^\star \neq \lambda_n^\star$, since no single $\bar{\lambda}$ can satisfy $\bar{\lambda} = \lambda_m^\star$ for all $m$ simultaneously. Minimizing over $\bar{\lambda} \in [0, 1]$ thus yields $\mathcal{J}_{uni} > \mathcal{J}_{adapt}$, completing the proof. $\square$

**Remark.** Our method performs anchor calibration via a logits-level regularizer rather than explicitly interpolating parameters. For theoretical tractability, we adopt a standard surrogate view that this calibration *implicitly pulls* the client solution toward the global anchor, which motivates the interpolation model in Eq. (A.10).

---

[5]In Sec. 2, this weight is denoted by $\alpha_m$; in this section we use $\lambda$ to denote the same intervention strength for analysis convenience, i.e., $\lambda \equiv \alpha$.

# B. Implementation Details

### B.1. Notation and Definition

For clarity and ease of reference, all notations and definitions used throughout this paper are summarized in Table B.1.

### B.2. Experimental Setup

Unless otherwise stated, we implement all algorithms using PyTorch on eight NVIDIA RTX 4090 GPUs. All methods utilize a ResNet-18 backbone tailored to the resolution of the respective datasets (e.g., $48 \times 48$ for FER2013). Local training runs for $E = 5$ epochs per communication round using the SGD optimizer with a learning rate of $\eta = 0.01$, momentum of $0.9$, and weight decay of $10^{-4}$. The local batch size is set to $B = 16$, and the total communication rounds are set to $T = 100$.

### B.3. Loss Function

Following the standard LDL paradigm, we employ the Kullback-Leibler (KL) divergence as the objective function to measure the discrepancy between the predicted label distribution and the ground truth. Specifically, for a client $u_m$ with dataset $\mathcal{D}_m = \{(\mathbf{x}_{m,j}, \mathbf{d}_{m,j})\}_{j=1}^{N_m}$, the local loss function is defined as:

$$\mathcal{L}_{LDL}(\mathbf{w}_m) = \frac{1}{N_m} \sum_{j=1}^{N_m} \sum_{c=1}^{C} d_{m,j}^{(c)} \ln \frac{d_{m,j}^{(c)}}{p_m^{t(c)}(\mathbf{x}_{m,j})} \tag{B.1}$$

where $d_{m,j}^{(c)}$ and $p_m^{t(c)}(\mathbf{x}_{m,j})$ denote the $c$-th element of the ground-truth distribution $\mathbf{d}_{m,j}$ and the predicted distribution $\mathbf{p}_m^t(\mathbf{x}_{m,j})$, respectively.

### B.4. Method-specific Hyperparameters

For our proposed FedQual framework, we set the modulation hyperparameters in the client-side rectifier (Eq. 3) as $\beta = 5$ and $\lambda_0 = 0.5$, with the normalization factor $\tau$ set to the maximum possible quality score (e.g., 10 for the 10-expert setting). For the server-side aggregation (Eq. 6), the inverse temperature parameter is set to $\gamma_{temp} = 1$ to control the sharpness of the reweighting mechanism.

# C. Benchmark Construction Details

To establish rigorous benchmarks for Fed-LDL, we selected four publicly available datasets widely recognized in the computer vision community, covering two core tasks: Facial Emotion Recognition (FER) and Image Quality Assessment (IQA). We further applied a systematic data cleaning procedure to exclude corrupted or invalid samples, thereby constructing a reliable benchmark suitable for federated evaluation.

### C.1. Source Datasets

**1. FER2013** This dataset originates from the ICML 2013 Challenges in Representation Learning (Goodfellow et al., 2015). Unlike datasets collected in controlled laboratory settings, FER2013 images were retrieved via the Google Image Search API, representing in-the-wild conditions. The raw data provides single-label annotations across seven basic emotions (Anger, Disgust, Fear, Happiness, Sadness, Surprise, and Neutral). We selected this dataset because its uncontrolled environment introduces significant occlusion, pose variation, and illumination interference, which naturally create semantic ambiguity suitable for label distribution learning.

**2. FI** This dataset is a large-scale benchmark for affective computing which consists of over 23,000 real-world images collected from social media platforms (Flickr and Instagram) (You et al., 2016). The original annotations define eight emotion categories based on Mikels' psychological model: Amusement, Anger, Awe, Contentment, Disgust, Excitement, Fear, and Sadness. Due to its social media origin, the FI dataset exhibits high content diversity and background complexity, making it an ideal testbed for simulating decentralized user data in federated learning scenarios.

**3. KADID-10k** The Konstanz Artificially Distorted Image quality Database (KADID-10k) contains 10,125 distorted images derived from 81 pristine reference images collected from Pixabay (Lin et al., 2019). All images are standardized to a resolution of $512 \times 384$ pixels. The dataset covers 25 distortion types (including blur, color shifts, and compression)

*Table B.1.* Mathematical Notations and Definitions

| Symbol | Definition |
|---|---|
| $M$ | The total number of clients in the Fed-LDL system. |
| $\mathcal{U}$ | The set of all participating clients. |
| $\mathcal{D}_m$ | The local dataset held by client $u_m$. |
| $N_m$ | The number of samples in the local dataset of client $m$. |
| $C$ | The number of classes (label space dimension). |
| $t$ | The communication round index. |
| $\mathbf{w}_m^t$ | The local model parameters of client $m$ at round $t$. |
| $\mathbf{z}_m^t(\mathbf{x})$ | The logits vector output by the model. |
| $\mathbf{p}_m^t(\mathbf{x})$ | The predicted label distribution, calculated as $\text{softmax}(\mathbf{z}_m^t(\mathbf{x}))$. |
| $q_m$ | Quality Indicator for client $m$, quantifying the statistical fidelity of $\mathcal{D}_m$. |
| $\mathcal{A}(\mathbf{x})$ | Global Semantic Anchor (GSA), defined as the logits of the global model $\mathbf{z}(\mathbf{x}; \mathbf{w}_g^t)$. |
| $\mathcal{R}(\cdot, \cdot)$ | The semantic alignment regularizer. |
| $\alpha_m$ | Client-specific calibration weight regulating the trade-off between local learning and anchor calibration. |
| $\tau$ | Normalization factor for $q_m$ (e.g., set to the maximum quality score). |
| $\beta$ | Hyperparameter controlling the sharpness of the transition in the calibration weight function. |
| $\lambda_0$ | An intervention offset parameter for the calibration weight. |
| $S_m$ | Effective reliable information provided by client $m$, defined as $S_m \triangleq N_m \cdot q_m$. |
| $\rho_t$ | Trust annealing factor at round $t$, controlling the transition from quality-aware to quantity-aware weighting. |
| $\widetilde{S}_m^t$ | Intermediate score for aggregation at round $t$, defined as $(S_m)^{1-\rho_t}(N_m)^{\rho_t}$. |
| $\gamma_{temp}$ | The inverse-temperature parameter controlling the sharpness of the weighting distribution. |
| $\sigma_m^2$ | Local variance, defined as $\mathbb{E}[\|\mathbf{w}_m^{\text{loc}} - \mathbf{w}_m^\star\|_2^2]$. |
| $\delta_m^2$ | Global shift, defined as $\|\mathbf{w}_g^t - \mathbf{w}_m^\star\|_2^2$. |
| $\rho_{noise}$ | The ratio of low-quality clients. |
| $K$ | Parameter for Top-K operation used to discretize distributions. |
| $\gamma$ | Concentration parameter of the Dirichlet distribution for partitioning. |
| $\rho_{online}$ | The ratio of activated clients per communication round. |

across 5 intensity levels. The original ground truths were obtained via a crowdsourced Degradation Category Rating (DCR) protocol with 30 ratings per image. We selected KADID-10k because its diverse distortion types effectively simulate the

hardware-induced quality variations (e.g., sensor noise, transmission artifacts) prevalent in federated edge devices.

**4. PIPAL** The Perceptual Image Processing ALgorithms (PIPAL) dataset is a large-scale IQA benchmark explicitly designed to evaluate Image Restoration (IR) algorithms (Gu et al., 2020). It contains 29,000 distorted images generated from 250 high-quality reference patches ($288 \times 288$ pixels) selected from the DIV2K and Flickr2K datasets. A distinguishing feature of PIPAL is its inclusion of 40 distortion types, specifically the outputs of varying IR algorithms, including GAN-based Super-Resolution methods (e.g., ESRGAN). These GAN-based distortions introduce novel "texture-like" noise that violates natural image statistics, posing a severe challenge to model robustness. Furthermore, the original labels were derived using the Elo rating system based on over 1.13 million human judgements, ensuring high-fidelity reliability for our benchmark construction.

## C.2. Annotation Protocol

Unlike previous datasets that rely on unmonitored crowdsourcing platforms, which often introduce significant label noise due to annotator irresponsibility or environmental variance, we adopted a rigorous Laboratory-Controlled Expert Ensemble protocol. This protocol ensures that the generated label distributions reflect genuine semantic ambiguity rather than annotation errors.

**Expert Recruitment and Demographics.** We recruited two fixed panels of 10 domain experts each, one specializing in Facial Emotion Recognition (FER) and the other in Image Quality Assessment (IQA), from university research laboratories with expertise in Computer Vision and Psychology. To ensure high-fidelity annotations and generalization, the panel construction adhered to strict criteria: (i) **Screening and Expertise:** All candidates passed the *Ishihara Color Vision Test* and demonstrated normal visual acuity. For IQA tasks, experts possess academic backgrounds in digital image processing, ensuring proficiency in distinguishing complex distortions (e.g., GAN artifacts). For FER tasks, experts received training based on the *Facial Action Coding System (FACS)* and were evaluated on micro-expression identification. (ii) **Diversity Protocol:** To minimize demographic bias, we enforced a strict diversity protocol. Each panel consists of an equal split of *5 males and 5 females*, which is crucial for FER tasks given potential gender differences in emotion perception. Furthermore, experts were selected from diverse regional backgrounds to mitigate cultural bias in interpreting facial expressions or aesthetic preferences.

**Training and Calibration Phase.** Prior to formal annotation, we conducted a mandatory calibration phase to align internal standards and minimize intra-rater variance. First, experts independently annotated a *Pilot Set* of 100 representative images covering high-ambiguity cases. Subsequently, we analyzed the results and held *Consensus Meetings* to resolve samples with high disagreement (e.g., distinguishing between "Sadness" and "Neutral" in low-intensity expressions). This process established a unified consensus on boundary cases. Finally, only experts who achieved a high consistency score against the group consensus during this pilot phase were admitted to the final panel.

**Controlled Environment and Ethics.** To eliminate environmental noise commonly observed in crowdsourcing, all annotations were conducted in a standardized laboratory environment. **Physical Setup:** Experts used calibrated professional monitors under uniform ambient lighting to ensure consistent color and contrast perception. **Fatigue Management:** Annotation sessions were strictly limited to 45 minutes with mandatory breaks to prevent visual fatigue from degrading data quality. **Compensation and Ethics:** All participants provided informed consent and were compensated at a rate significantly higher than the local minimum hourly wage, acknowledging the high cognitive load of the expert annotation task.

## C.3. Recruitment and Consent Form for FER Task

Below is the template of the recruitment and informed consent form provided to the 10 domain experts who participated in the Facial Emotion Recognition (FER) annotation study.

## Participant Recruitment Form for Visual Emotion Annotation Study

**Study Overview**
We are conducting research on Federated Label Distribution Learning (Fed-LDL). The goal is to collect high-quality label distribution annotations from human experts to support the development of a robust and trustworthy Federated Label Distribution Learning (Fed-LDL) framework.

**Participation Details**

- **Task:** Participants will perform Cognitive-Guided Multi-Label Annotation. For each image, you must identify all applicable emotions from a set of 8 keywords based on facial expressions, body language, and scene context.

- **Duration:** Each annotation session will take approximately 45 minutes, with mandatory breaks to prevent visual and cognitive fatigue.

- **Compensation:** Participants will receive **$100 USD** upon completion of the task.

- **Eligibility:** Participants must pass the Ishihara Color Vision Test, possess a background in CV or Psychology, and have received training in the Facial Action Coding System (FACS).

**Ethics and Data Use**

- Participation is entirely voluntary. You may withdraw at any time without penalty.

- All responses and personal data will be anonymized and used exclusively for academic research.

- The resulting datasets will be released under an academic license for non-commercial use.

**Consent Declaration**
By signing below, I acknowledge that I have read the above information and voluntarily agree to participate in this study. I understand I may withdraw at any time and that my responses will remain confidential.

Name (print): ______________________     Signature: ______________________     Date: ______________

**Contact Information:**
Principal Investigator: Dr. XXX
Email: XXXX@XXXX.edu
Phone: +XX-XXXX-XXXX

### C.4. Recruitment and Consent Form for IQA Task

Below is the template of the recruitment and informed consent form provided to the 10 domain experts who participated in the Image Quality Assessment (IQA) annotation study.

**Participant Recruitment Form for Image Quality Assessment Study**

**Study Overview**
We are conducting research on Federated Label Distribution Learning (Fed-LDL). The goal is to collect high-quality label distribution annotations from human experts to support the development of a robust and trustworthy Federated Label Distribution Learning (Fed-LDL) framework.

**Participation Details**

- **Task:** Participants will evaluate images across six perceptual dimensions: Sharpness, Noise, Exposure, Color Accuracy, Artifacts, and Stability, using a standardized 5-point scale.

- **Duration:** Annotation sessions are limited to 45 minutes to maintain high consistency in visual judgment.

- **Compensation:** Participants will receive **$100 USD** upon completion of the task.

- **Eligibility:** Participants must pass the Ishihara Color Vision Test and possess expertise in digital image processing to distinguish complex distortions such as GAN-based artifacts.

**Ethics and Data Use**

- Participation is entirely voluntary. You may withdraw at any time without penalty.

- All responses and personal data will be anonymized and used exclusively for academic research.

- The resulting datasets will be released under an academic license for non-commercial use.

**Consent Declaration**
By signing below, I acknowledge that I have read the above information and voluntarily agree to participate in this study. I understand I may withdraw at any time and that my responses will remain confidential.

Name (print): ______________________    Signature: ______________________    Date: ______________

**Contact Information:**
Principal Investigator: Dr. XXX
Email: XXXX@XXXX.edu
Phone: +XX-XXXX-XXXX

## C.5. Annotation Guidelines and Label Distribution Generation

To ensure consistent and high-quality supervision, we established standardized annotation guidelines for both tasks. The detailed scoring rubrics for Image Quality Assessment (IQA) and the annotation guidelines for Facial Emotion Recognition (FER) are illustrated in Figure C.1 and Figure C.2, respectively. Below, we detail the mathematical generation process of the ground-truth label distributions $\mathbf{d}_{GT}$ from these expert annotations.

**IQA: Multi-Dimensional Evaluation to Distribution.** To mitigate the subjectivity bias inherent in holistic scoring, we adopt a decomposed evaluation protocol as visualized in Figure C.1. Experts are required to evaluate images across six distinct perceptual dimensions(Sharpness, Noise, Exposure, Color Accuracy, Artifacts, and Stability) using a standardized 5-point scale. To construct the latent ground-truth distribution $\mathbf{d}_{GT}$, we preserve the diversity of expert opinions rather than enforcing a single consensus label. Specifically, for a given image, we first calculate an individual scalar score $s_m \in [1, 5]$ for each expert $m$ ($m = 1, \ldots, 10$) by averaging their ratings across the six dimensions. Subsequently, to capture the inherent ambiguity, each expert's score $s_m$ is converted into a "soft vote" vector $\mathbf{v}_m$ using linear interpolation. A score $s_m$ contributes a weight of $1 - |s_m - l|$ to the integer quality level $l$ if $|s_m - l| < 1$, and 0 otherwise. For instance, an expert score of $s_m = 3.2$ contributes 0.8 to the "Fair" class (Level 3) and 0.2 to the "Good" class (Level 4). Finally, the soft votes from all 10 experts are summed and normalized to generate the label distribution $\mathbf{d}_{GT} \in \Delta^5$. This process ensures that the

final distribution reflects both the central tendency of the visual quality and the variance in expert judgments.

| Evaluation Dimensions | Score 1 (Bad) | Score 2 (Poor) | Score 3 (Fair) | Score 4 (Good) | Score 5 (Excellent) |
|---|---|---|---|---|---|
| **Sharpness / Detail** | Severe blur, focus failure, or strong motion blur. | Overall soft; slight misfocus or motion trail; outlines visible but texture lost. | Generally clear but details are soft; minor local blur visible. | Mostly clear; subject and texture are good; slight softness or minor over-sharpening. | Natural sharpness; rich details; no over-sharpening halos or ringing. |
| **Noise/Grain** | Significant strong noise or severe smoothing (waxy artifact). | Obvious uniform noise affecting detail recognition; or texture loss due to denoising. | Visible noise but does not significantly affect subject recognition. | Slight fine grain; structure and texture well-preserved; no excessive denoising artifacts. | Clean and natural; fine grain; no smoothing artifacts or color blotches. |
| **Exposure / Dynamic Range** | Severe over/under-exposure; massive detail loss (clipping/crushed blacks). | Obvious exposure bias; significant detail loss in highlights or shadows. | Usable but range compressed; detail loss in one extreme (highlight/shadow). | Balanced exposure; minor clipping at extremes but subject hierarchy preserved. | Full dynamic range; balanced brightness; no obvious clipping or crushed blacks. |
| **Color / White Balance** | Severe color cast causing skin tone or neutral gray distortion. | Moderate cast (warm/cool/green); requires obvious manual correction. | Slight cast or minor saturation imbalance; still acceptable. | Generally natural; negligible local hue/saturation fluctuation. | Accurate skin tones and neutrals; excellent saturation and hue balance. |
| **Artifacts / Compression** | Severe blocking, ringing, banding, moiré, or interpolation traces. | Clearly visible artifacts (e.g., blocks/banding) interfering with key details. | Minor artifacts present; distinguishable but not distracting. | Mostly clean; traces (e.g., compression) visible only upon zooming. | Imperceptible artifacts; texture edges are clean and natural. |
| **Motion/Shake /RS** | Severe trails, shake, or rolling shutter distortion impacting subject. | Moderate shake or motion blur; obvious detail smearing. | Slight shake or rolling shutter distortion; subject remains recognizable. | Stable; no significant motion blur; negligible signs of instability. | Perfectly stable; no trails, shake, or geometric distortion. |

*Figure C.1.* Detailed Annotation Rubric for the IQA Task. Experts evaluate images across six perceptual dimensions (Sharpness, Noise, Exposure, Color, Artifacts, and Stability) on a 5-point scale ranging from 1 (Bad) to 5 (Excellent). The specific criteria for each grade serve as a standardized reference to minimize subjective variance and ensure objective quantification of image quality.

**FER: Cognitive-Guided Multi-Label Annotation.** To capture the inherent ambiguity of facial expressions, we employ a Cognitive-Guided Protocol as illustrated in Figure C.2. Unlike traditional single-label annotation, experts are instructed to analyze a comprehensive set of visual cues—including facial expressions, body language, posture, scene context, objects, colors, lighting, and composition—and subsequently select **all** applicable emotions from a predefined set of eight keywords (Anger, Disgust, Fear, Happiness, Sadness, Surprise, Neutral, Contempt). This multi-label approach allows for the characterization of complex, mixed emotional states often present in real-world imagery.

To transform these discrete multi-label annotations into a continuous label distribution $\mathbf{d}_{GT}$, we aggregate the independent judgments of the expert panel. Let $\mathcal{E} = \{e_1, \ldots, e_8\}$ denote the set of emotion categories. For a specific image, let $v_{m,k} \in \{0, 1\}$ represent the binary vote of the $m$-th expert ($m = 1, \ldots, 10$) for the $k$-th emotion category. We first compute an aggregated vote vector $\mathbf{v} = [N_1, \ldots, N_8]$, where $N_k = \sum_{m=1}^{10} v_{m,k}$ represents the total endorsement count for emotion $k$. The final ground-truth distribution $\mathbf{d}_{GT} \in \Delta^8$ is then derived via $L_1$ normalization:

$$\mathbf{d}_{GT} = \frac{\mathbf{v}}{\|\mathbf{v}\|_1 + \epsilon} \tag{C.1}$$

where $\epsilon$ is a small constant for numerical stability. This resulting dense distribution effectively quantifies the intensity and mixture of the perceived emotions.

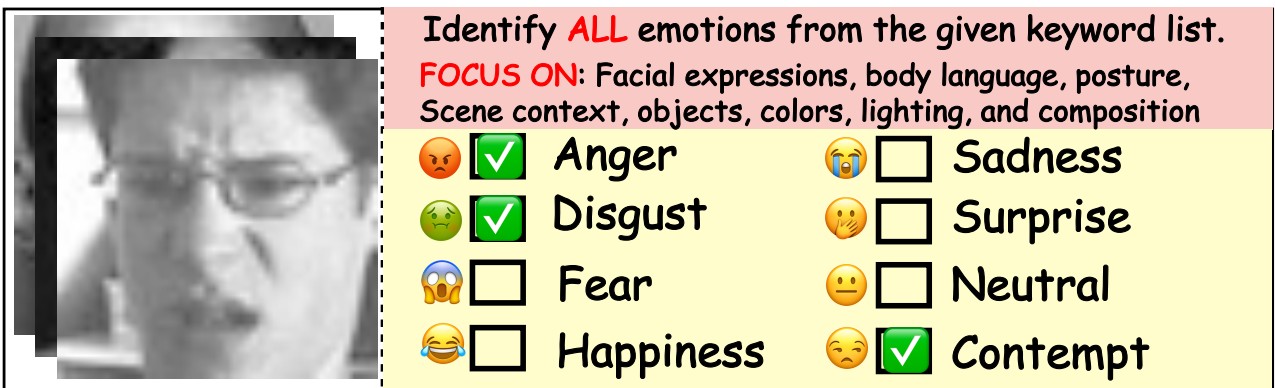

*Figure C.2.* Facial Emotion Recognition (FER) annotation guideline. For each image, annotators select all applicable emotions from a predefined set of eight emotion keywords based on the facial expressions present in the image, allowing multi-label emotion annotation.

## D. Complete Experimental Results

### D.1. Complete Ablation Studies

Supplementing the analysis in Section 6.2 of the main text, we present the comprehensive ablation results across all four benchmarks in Table D.1.

The full results corroborate the necessity of the dual-mechanism design. While the individual contribution of the local module (+A) may vary slightly depending on the dataset characteristics, the complete FedQual framework (+A+B) consistently achieves superior performance across the majority of metrics on all datasets. This empirical evidence confirms that the both components are essential for handling diverse scenarios of annotation quality heterogeneity.

*Table D.1.* Complete ablation study results across all four benchmarks. "Base" denotes the standard FedAvg. "+A" incorporates the *Quality-Modulated Rectifier*. "+A+B" represents the full framework. The green background highlights the local module, and the blue background highlights the full model. **Bold** indicates the best performance.

| Metric | FER-LDL | | | FI-LDL | | | KADID-LDL | | | PIPAL-LDL | | |
|---|---|---|---|---|---|---|---|---|---|---|---|---|
| | Base | +A | +A+B | Base | +A | +A+B | Base | +A | +A+B | Base | +A | +A+B |
| KL ↓ | 0.342 | 0.264 | **0.253** | 0.423 | 0.482 | **0.409** | 0.125 | 0.116 | **0.091** | 0.252 | 0.154 | **0.116** |
| Cheb ↓ | 0.213 | 0.174 | **0.172** | 0.228 | 0.235 | **0.217** | 0.144 | 0.145 | **0.120** | 0.168 | 0.166 | **0.139** |
| Clark ↓ | 1.323 | 1.281 | **1.260** | 1.858 | 1.869 | **1.855** | 1.195 | **0.992** | 1.068 | 1.398 | 1.399 | **1.391** |
| Canb ↓ | 3.097 | 2.951 | **2.887** | 4.513 | 4.543 | **4.480** | 2.717 | 2.641 | **2.572** | 2.510 | 2.481 | **2.387** |
| Inter ↑ | 0.696 | 0.739 | **0.745** | 0.667 | 0.653 | **0.679** | 0.853 | 0.852 | **0.877** | 0.816 | 0.814 | **0.844** |
| Cos ↑ | 0.816 | 0.868 | **0.874** | 0.810 | 0.802 | **0.817** | 0.957 | 0.956 | **0.967** | 0.929 | 0.937 | **0.958** |

### D.2. Robustness to Annotation Quality Heterogeneity

We here present the comprehensive visualization of FedQual's robustness against the dual dimensions of annotation quality degradation across all four benchmarks.

Figure D.1 details the performance trends as the quality indicator $q_m$ varies from 5 to 8. Consistent with the main text, the curves remain remarkably flat across all datasets. This invariance across diverse visual tasks serves as strong empirical evidence that our framework effectively mitigates local optimization drift, preserving model fidelity regardless of the local

noise severity. Furthermore, Figure D.2 illustrates the comprehensive metric breakdown as the prevalence of low-quality clients increases from 25% to 75%. Despite the potential for noise domination, FedQual exhibits exceptional stability across all evaluation metrics. These findings confirm that FedQual is robust to both localized label noise and global client imbalance.

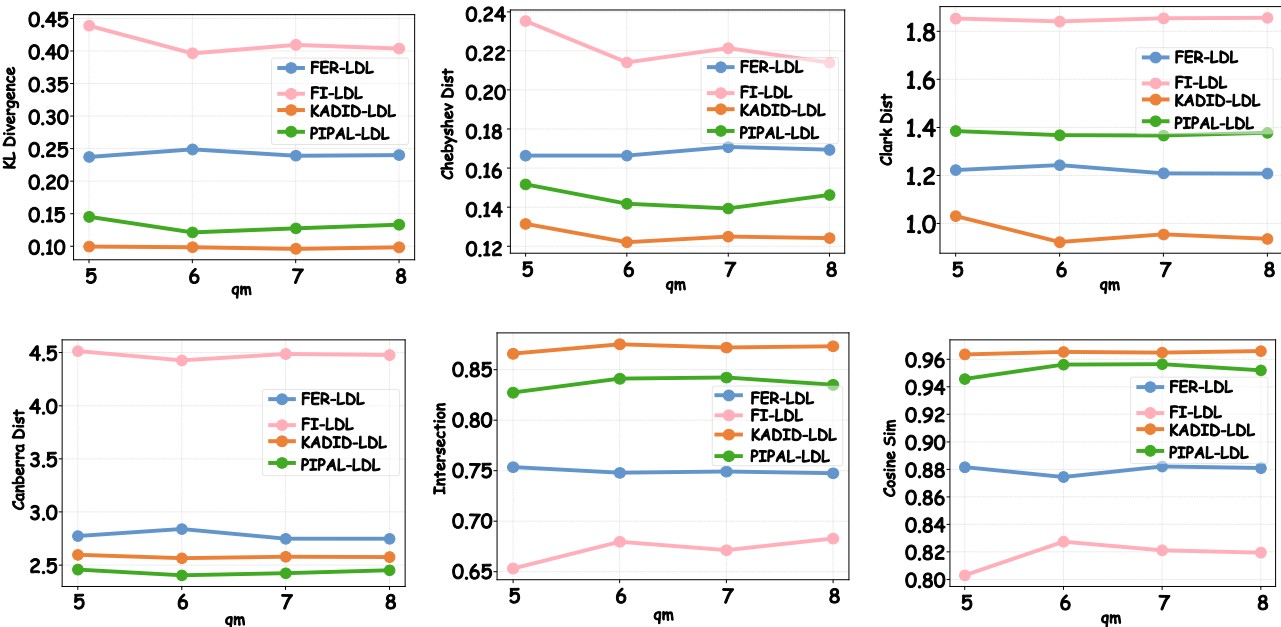

*Figure D.1.* **Sensitivity to Noise Intensity:** The impact of varying the quality indicator $q_m \in \{5, 6, 7, 8\}$ on FER-LDL, FI-LDL, KADID-LDL, and PIPAL-LDL. The flat curves indicate that FedQual maintains high performance regardless of the local noise variance.

### D.3. Robustness to Label Distribution Skew

To complement the analysis in the main text, we present the complete sensitivity analysis regarding the label distribution skew parameter $\gamma$ across all four benchmarks (FER-LDL, FI-LDL, PIPAL-LDL, and KADID-LDL). As illustrated in Figure D.3, we vary $\gamma \in \{0.25, 0.5, 0.75, 1.0\}$. Consistent with the observations on FER-LDL, FedQual demonstrates high stability across all datasets and metrics. These extensive results further corroborate the robustness of our framework against varying degrees of non-IID label skew in diverse vision tasks.

### D.4. Robustness to Label Multiplicity

We extend the analysis of label multiplicity robustness to all four benchmarks in Figure D.4. By varying the Top-$K$ parameter ($K \in \{1, 2, 3, 4\}$) used for constructing non-IID partitions, we observe that the stability reported in the main text is not specific to FER-LDL but generalizes to FI-LDL, PIPAL-LDL, and KADID-LDL. The consistently flat trends across all metrics confirm that FedQual effectively handles varying degrees of label sparsity and feature skew, regardless of the specific task domain.

### D.5. Scalability to Federation Size

We present the complete scalability analysis across all four benchmarks in Figure D.5. By expanding the evaluation range of the total client pool size ($\{25, 50, 75, 100\}$) to FER-LDL, PIPAL-LDL, and KADID-LDL, we observe consistent stability in performance, mirroring the results on FI-LDL reported in the main text. These extensive experiments confirm that FedQual can effectively scale to larger federations without suffering from performance degradation.

## D.6. Robustness to Partial Participation

We extend the partial participation analysis to all four constructed benchmarks in Figure D.6. By varying the active client ratio $\rho_{online} \in \{0.2, 0.4, 0.6, 0.8\}$, we observe that the high stability reported in the main text is consistent across both emotion recognition and IQA tasks. These results further verify that FedQual remains effective in practical scenarios with intermittent client availability.

# E. Additional Discussion

## E.1. Limitations and Future Directions

FedQual currently simplifies the quality indicator $q_m$ by associating it with the number of annotators. In real-world scenarios, annotation fidelity is often decoupled from quantity; for instance, a single domain expert may provide a more precise and nuanced label distribution than a consensus of multiple non-expert annotators. Future research could explore unsupervised quality estimation techniques that infer client reliability directly from the local data manifold or the consistency of learning dynamics, thereby reducing the reliance on external metadata for quality assessment.

Another limitation concerns the trust relationship regarding quality reporting in privacy-sensitive environments. Our framework assumes that quality metadata can be truthfully acquired, yet in practice, verifying the authenticity of such indicators without inspecting raw data remains a challenge. Malicious participants might misreport their quality scores to gain disproportionate influence over the global model, leading to potential security vulnerabilities. Integrating privacy-preserving verification protocols, such as zero-knowledge proofs or hardware-based trusted execution environments, to validate the integrity of the annotation process without data leakage presents a promising direction for enhancing federated robustness.

## E.2. Reproducibility

We provide a comprehensive description of our FedQual framework and its quality-adaptive training pipeline in Section 3, with supporting theoretical analysis of the adaptive calibration mechanism documented in Section 4. Detailed implementation aspects, including the federated setup (ResNet-18 backbone), the Global Semantic Anchor (GSA) calibration function, and method-specific hyperparameters such as $\beta$ and $\lambda_0$, are meticulously documented in Appendix B. All experiments were conducted using PyTorch on NVIDIA RTX 4090 GPUs , utilizing standardized communication protocols ($T = 100$ rounds) and fixed random seeds to eliminate variance from arbitrary initializations. To facilitate further research and ensure full reproducibility, we will release our source code, training scripts, and the four newly constructed Fed-LDL benchmarks (FER-LDL, FI-LDL, PIPAL-LDL, and KADID-LDL). This release allows the community to verify our results and adapt the FedQual framework to other privacy-sensitive applications requiring nuanced semantic understanding.

## E.3. Use of LLMs in Writing

Large language models (LLMs) was used solely for language polishing, enhancing clarity, grammar, and stylistic consistency. The LLM was not involved in idea generation, method design, algorithm development, experimental planning, or result analysis. All scientific contributions, including the formulation of research questions, experimental design, data analysis, and interpretation of results, were made by the authors themselves.

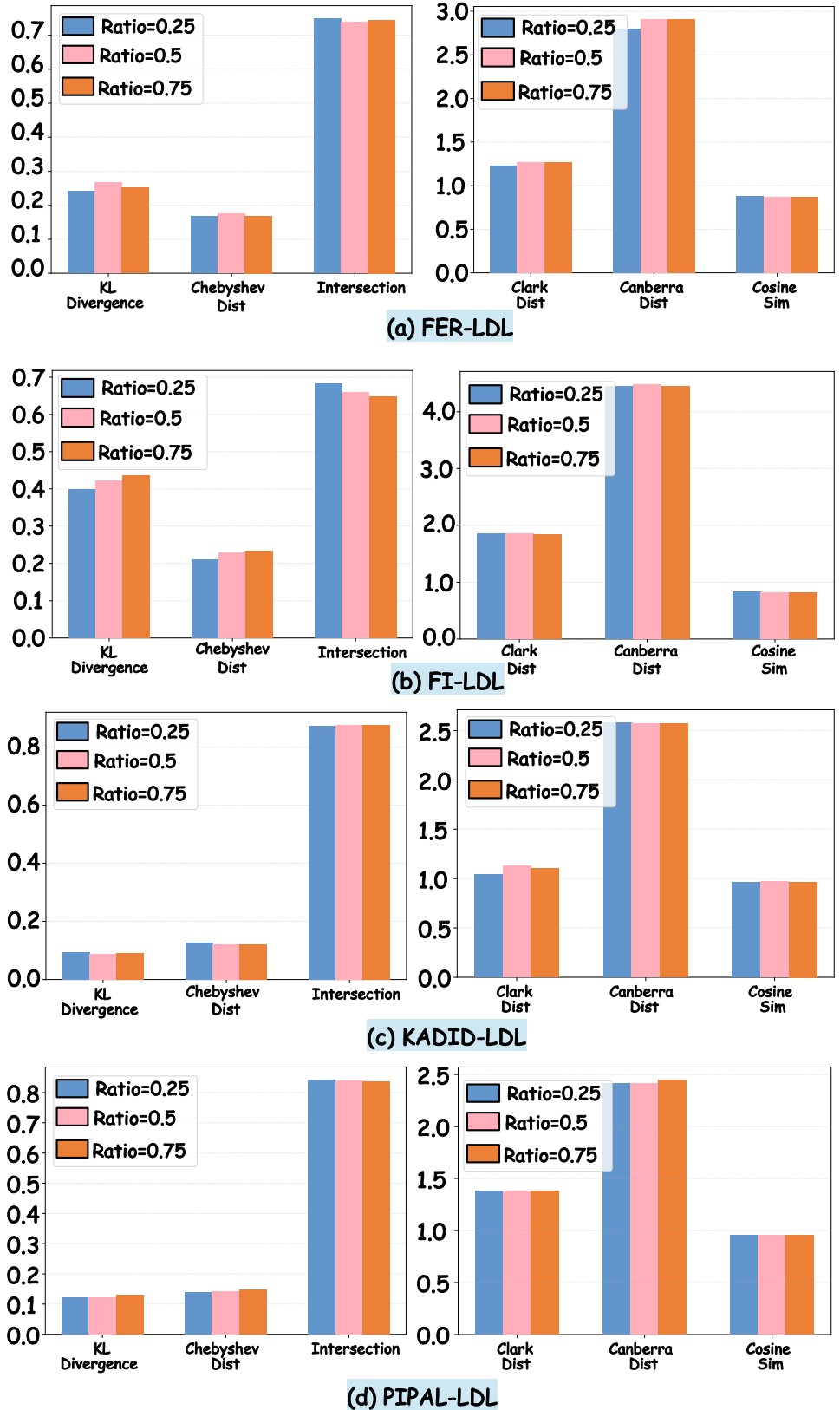

*Figure D.2.* **Impact of Noisy Client Ratio:** Performance comparison under different ratios of low-quality clients ($\rho_{noise} \in \{0.25, 0.50, 0.75\}$). FedQual exhibits strong resistance to noise domination, sustaining stable metrics even when 75% of clients are noisy.

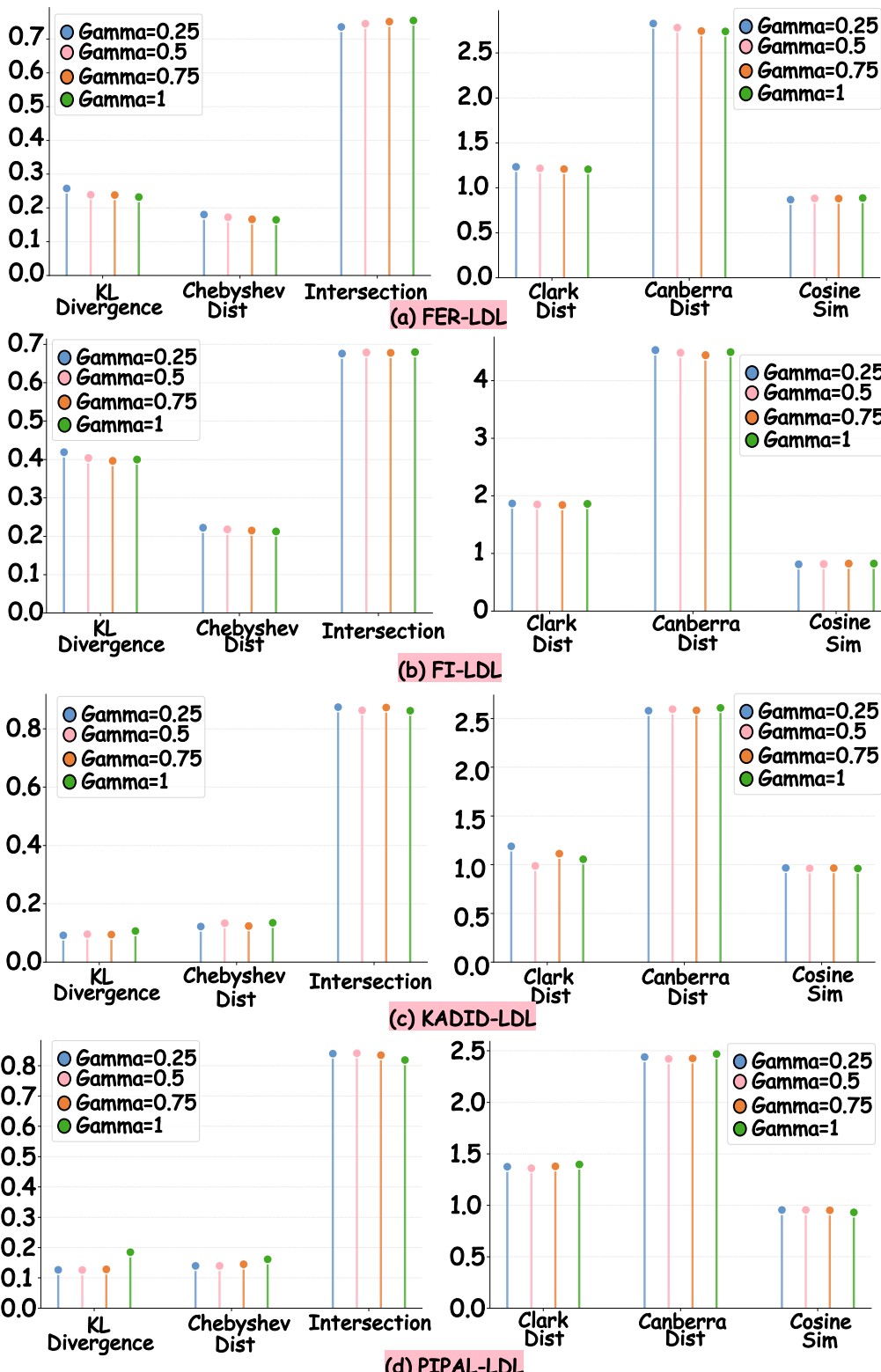

*Figure D.3.* Sensitivity analysis of the label distribution skew parameter $\gamma$ on the four benchmark. We evaluate the performance across varying skew levels $\gamma \in \{0.25, 0.5, 0.75, 1.0\}$. The results across six metrics demonstrate that our proposed FedQual framework remains robust to different degrees of label distribution skew.

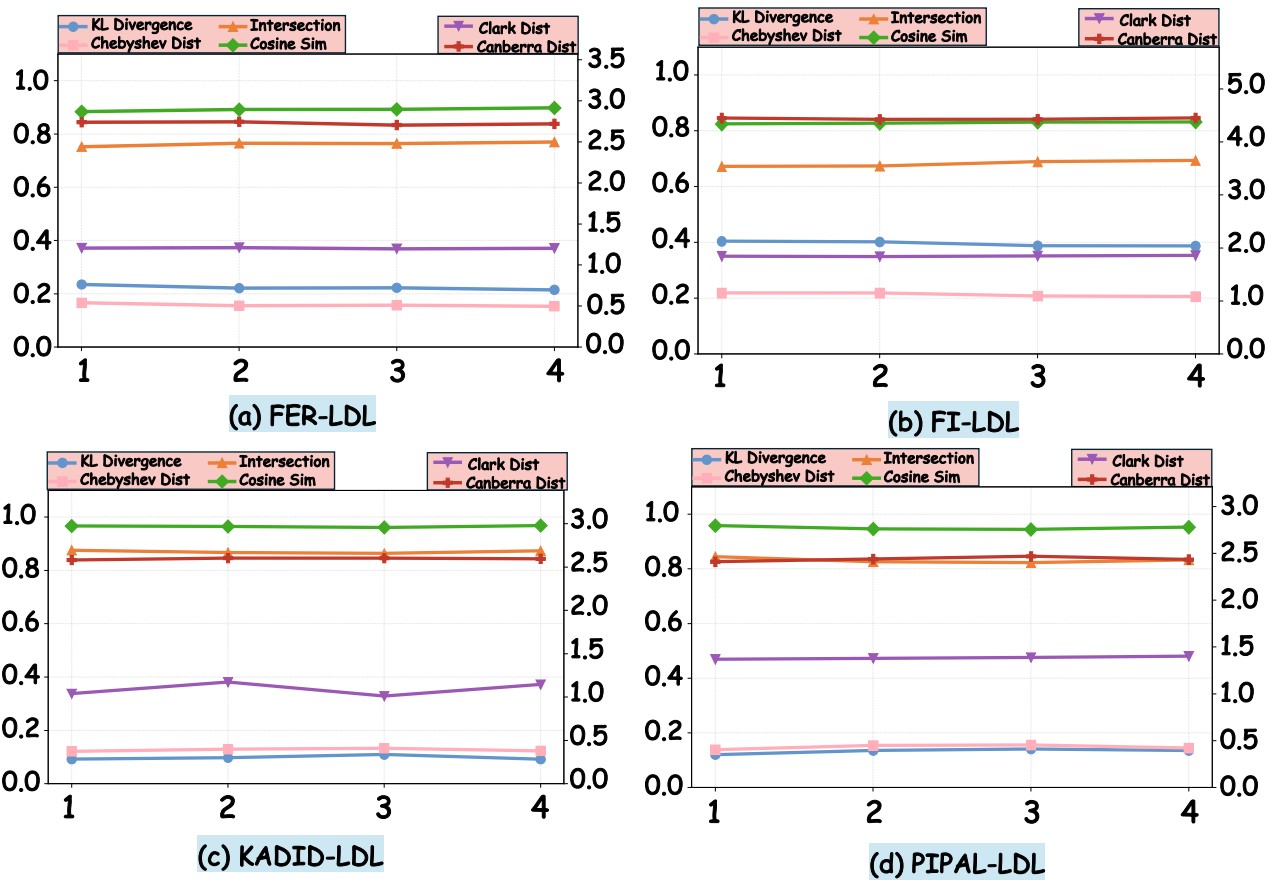

*Figure D.4.* Impact of partition label multiplicity (Top-$K$) on the four benchmark. We vary the Top-$K$ parameter ($K \in \{1, 2, 3, 4\}$) used to discretize $\mathbf{d}_{GT}$ for simulating Non-IID feature skew. The flat trends across all six metrics demonstrate that FedQual maintains robust performance regardless of the label density used for client partitioning.

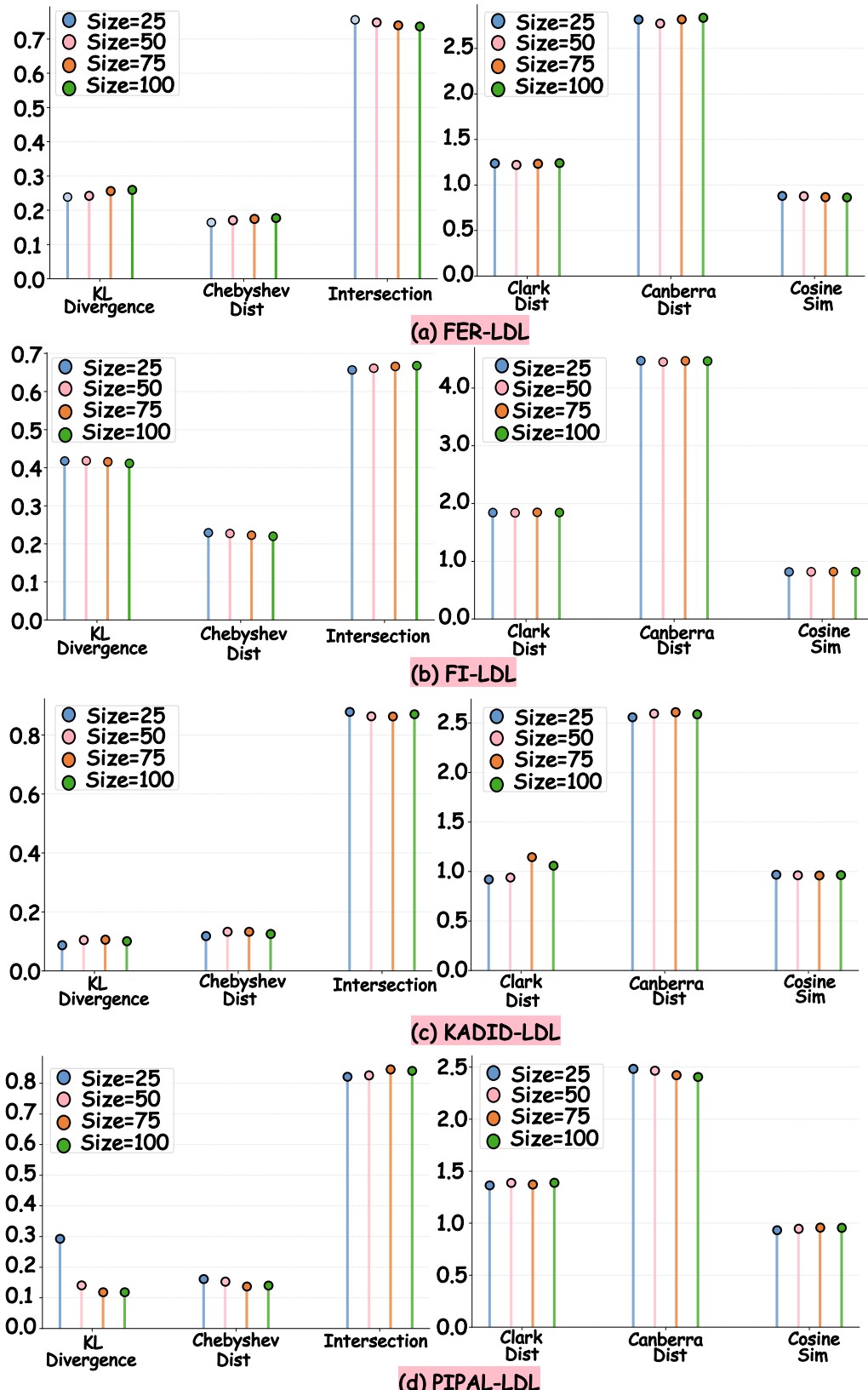

*Figure D.5.* Scalability analysis regarding the federation size on the four benchmark. We evaluate the model performance by varying the total client pool size in the range of $\{25, 50, 75, 100\}$. The consistent results across six metrics demonstrate that FedQual scales effectively to larger federations without performance degradation.

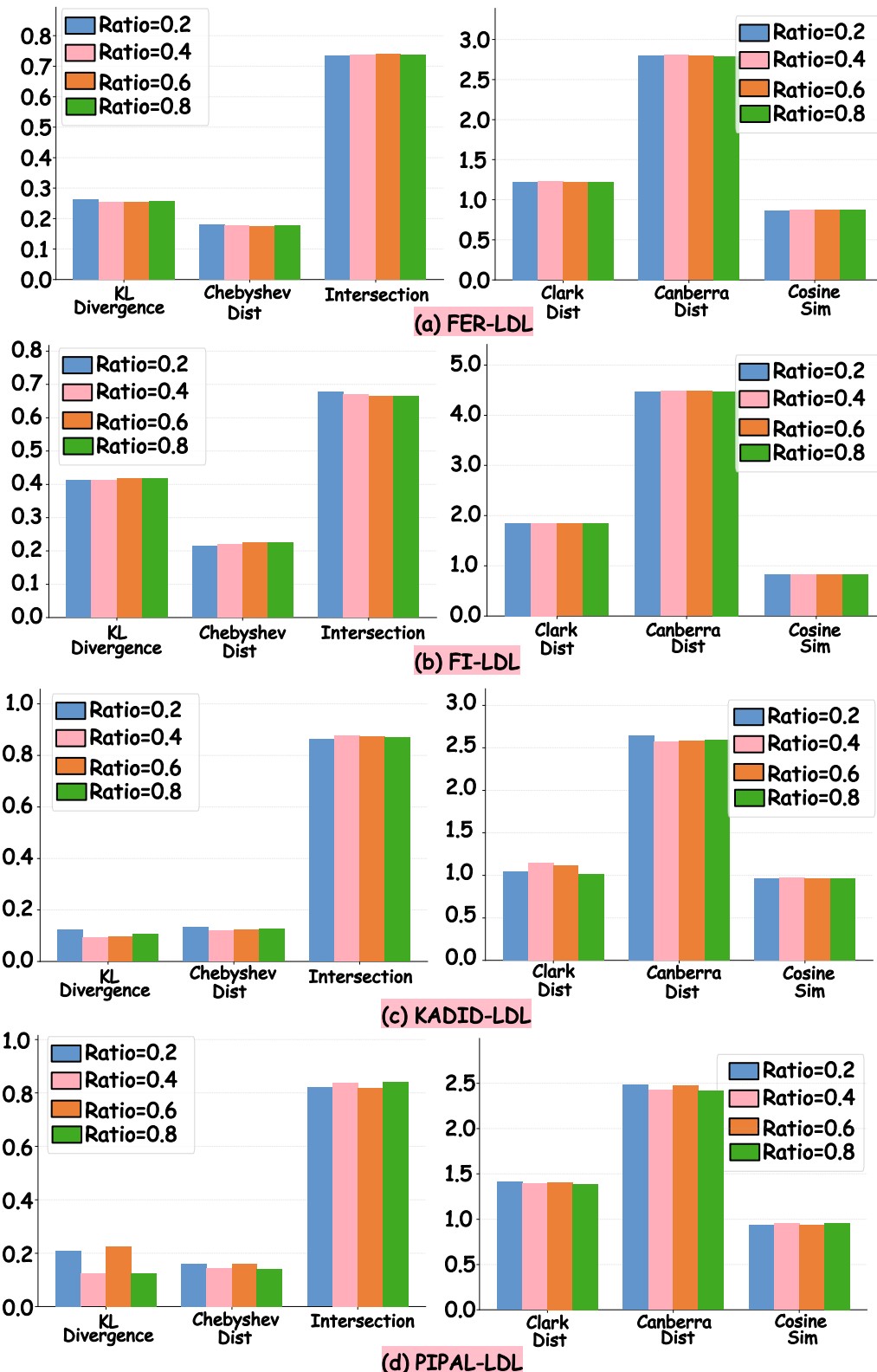

*Figure D.6.* Robustness analysis regarding the client participation ratio on the four benchmarks. We evaluate model performance by varying the ratio of active clients per round in the range of $\{0.2, 0.4, 0.6, 0.8\}$. The consistent performance across all six metrics demonstrates that our framework remains effective even under low participation rates.

