# OpenReview forum: "Trustworthy Federated Label Distribution Learning under Annotation Quality Disparity"
_ICML.cc/2026/Conference — ICML 2026 regular_

### Official Review · Reviewer_afAv · 2026-03-09

**Soundness:** 3
**Presentation:** 3
**Significance:** 4
**Originality:** 4
**Overall Recommendation:** 5
**Confidence:** 4

**Summary:**

This paper looks at a fairly natural but still under-addressed problem in Fed-LDL: once supervision quality differs across clients, standard sample-size-based aggregation can become unreliable. To handle this, the paper proposes FedQual, which combines a client-side quality modulated rectification mechanism with a server-side quality-aware aggregation rule that weights clients by effective reliable information rather than raw data volume. The paper also builds four new Fed-LDL benchmarks and includes a theorem arguing that client-specific
calibration is strictly preferable to a uniform calibration strength under heterogeneous supervision quality.

**Compliance With Llm Reviewing Policy:**

Affirmed.

**Final Justification:**

The author has thoroughly addressed my concerns. Therefore, I remain positive attitude.

**Key Questions For Authors:**

1. I'm curious about the demographics of the 10-expert panels. Did you implement any specific strategies during recruitment to ensure the ground-truth distributions aren't biased toward a single cultural or regional perspective?

2. How sensitive is the framework to the GSA hyperparameters? Do they work out-of-the-box across different tasks, or do they require heavy tuning? A brief discussion or a small sensitivity table would really help demonstrate the method's robustness.

**Limitations:**

Yes

**Strengths And Weaknesses:**

Strengths:

1. The proposed method aligns well with the problem and is intuitively reasonable.

2. The empirical evaluation is quite comprehensive, and the experimental results are encouraging.

3. The authors not only provide theoretical guarantees but also introduce several relevant datasets to fill gaps in the field. Therefore, the contributions of this paper are sufficient.

Weakness:

1. The paper seems to rely on fixed random seeds for the reported results. It would be more convincing to see a small note on run-to-run variance, or results averaged over a few runs, just to show the gains are not too seed-sensitive.

2. It appears that the authors have not provided a link to the datasets. Could the datasets be made publicly available/open-sourced?

---

> ### Author Rebuttal · Authors · 2026-03-28
>
> **Response to Reviewer afAV.**
>
> We sincerely appreciate the reviewer’s valuable feedback and constructive comments. Below, we address each concern in turn.
>
> > Q1(weakness 1): The paper seems to rely on fixed random seeds for the reported results. It would be more convincing to see a small note on run-to-run variance, or results averaged over a few runs.
>
> Thank you for this suggestion. In the main paper, all reported results use the default random seed 0. To further address the concern about seed sensitivity, we repeated the experiments with **three random seeds** (0,10,100) on FER-LDL and PIPAL-LDL, and report the **mean $\pm$ std** below. We will add this multi-seed note in the final version for completeness.
>
> | Dataset | Method | KL Divergence $\downarrow$ | Intersection $\uparrow$ | Cosine Sim $\uparrow$ |
> |---|---|---|---|---|
> | FER-LDL | FedQual | $\mathbf{0.2485 \pm 0.0048}$ | $\mathbf{0.7454 \pm 0.0013}$ | $\mathbf{0.8767 \pm 0.0030}$ |
> | FER-LDL | FedQRct | $0.2657 \pm 0.0019$ | $0.7354 \pm 0.0027$ | $0.8653 \pm 0.0023$ |
> | FER-LDL | FedProx | $0.2987 \pm 0.0075$ | $0.7158 \pm 0.0033$ | $0.8432 \pm 0.0047$ |
>
> | Dataset | Method | KL Divergence $\downarrow$ | Intersection $\uparrow$ | Cosine Sim $\uparrow$ |
> |---|---|---|---|---|
> | PIPAL-LDL | FedQual | $\mathbf{0.1225 \pm 0.0065}$ | $\mathbf{0.8394 \pm 0.0049}$ | $\mathbf{0.9535 \pm 0.0045}$ |
> | PIPAL-LDL | FedQAgg | $0.2348 \pm 0.1769$ | $0.8273 \pm 0.0110$ | $0.9437 \pm 0.0131$ |
> | PIPAL-LDL | FedProx | $0.1303 \pm 0.0026$ | $0.8350 \pm 0.0015$ | $0.9370 \pm 0.0021$ |
>
> From the above results, we draw two main observations:
>
> - **FedQual is itself stable across runs.** On both FER-LDL and PIPAL-LDL, FedQual shows small run-to-run variance, indicating that it is not overly sensitive to random initialization.
>
> - **The main conclusion does not depend on a single lucky seed.** After averaging over three runs, FedQual still achieves the best performance among the compared methods on both representative benchmarks, so the overall conclusion of the paper remains unchanged under multi-seed evaluation.
>
> > Q2(weakness 2):It appears that the authors have not provided a link to the datasets. Could the datasets be made publicly available/open-sourced?
>
> Thank you for the suggestion. Yes, we plan to publicly release the benchmark artifacts. We will provide the **dataset release information** and **the project website** in the final version. Please also see our response to **Reviewer Qeqz (Q2)** for more details on the release plan.
>
> > Q3(key question 1): Did you implement any specific strategies during recruitment to ensure the ground-truth distributions aren't biased toward a single cultural or regional perspective?
>
> Thank you for this important question. Yes, we explicitly considered this issue during recruitment and benchmark construction.
>
> - **Demographic diversity in panel composition.**
>   For both the FER and IQA panels, we recruited annotators from diverse regional backgrounds and maintained an equal gender split (5 male / 5 female per panel), to reduce over-reliance on a single demographic perspective.
>
> - **Task-specific screening.**
>   We used task-specific screening to ensure that the recruited annotators had the background needed for the corresponding benchmark.
>
> - **Calibration before formal annotation.**
>   Before formal labeling, we conducted a pilot calibration stage with consensus meetings to align annotation criteria and reduce systematic inconsistency across annotators.
>
> We will clarify this more explicitly in the revised version, and more details are provided in **Appendix D**.
>
> > Q4(key question 2): How sensitive is the framework to the GSA hyperparameters? Do they work out-of-the-box across different tasks, or do they require heavy tuning?
>
> Thank you for this helpful suggestion. We agree that a small sensitivity analysis helps clarify whether the GSA-based local calibration requires heavy tuning. To directly test this, we performed a controlled study in which we fixed the anchor-regularization strength $\alpha_m$ to several values, and evaluated the resulting performance on FER-LDL and PIPAL-LDL.
>
> **(1) FER-LDL**
> | $\alpha_m$ | KL Divergence $\downarrow$ | Intersection $\uparrow$ | Cosine Sim $\uparrow$ |
> |---|---|---|---|
> | 0.10 | 0.2491 | 0.7428 | 0.8766 |
> | 0.25 | 0.2521 | 0.7453 | 0.8749 |
> | 0.50 | 0.2719 | 0.7362 | 0.8601 |
> | 0.75 | 0.2478 | 0.7443 | 0.8777 |
> | 1.00 | **0.2425** | **0.7504** | **0.8805** |
>
> **(2) PIPAL-LDL**
> | $\alpha_m$ | KL Divergence $\downarrow$ | Intersection $\uparrow$ | Cosine Sim $\uparrow$ |
> |---|---|---|---|
> | 0.10 | 0.1284 | 0.8385 | 0.9532 |
> | 0.25 | 0.1325 | 0.8341 | 0.9498 |
> | 0.50 | **0.1274** | **0.8415** | **0.9538** |
> | 0.75 | 0.1464 | 0.8182 | 0.9380 |
> | 1.00 | 0.1292 | 0.8363 | 0.9529 |
>
> These results show that the framework is not sensitive to the exact value of $\alpha_m$. This further suggests that the GSA-based calibration is robust in practice.

---

> > ### Author Rebuttal · Reviewer_afAv · 2026-04-02
> >
> > Thanks for the detailed responses. My concerns have been fully addressed. I would like to keep my score.

---

> > > ### Author Response · Authors · 2026-04-05
> > >
> > > Dear Reviewer afAv,
> > >
> > > We sincerely appreciate your thoughtful evaluation and are delighted that our rebuttal has addressed your concerns.
> > >
> > > Best regards,
> > >
> > > Authors

---

### Official Review · Reviewer_2NNQ · 2026-03-13

**Soundness:** 4
**Presentation:** 4
**Significance:** 4
**Originality:** 4
**Overall Recommendation:** 4
**Confidence:** 4

**Summary:**

This paper studies a practical but still underexplored issue in Federated Label Distribution Learning (Fed-LDL): different clients may have very different supervision quality, so local updates are not equally trustworthy, while standard aggregation methods tend to overemphasize large but noisy clients. To address this, this paper proposes FedQual. On the client side, FedQual introduces a Global Semantic Anchor to perform quality-adaptive calibration. On the server side, instead of relying only on sample size, the method uses a quality-aware aggregation strategy built on “effective reliable information” and gradually shifts from emphasizing quality to taking both quality and quantity into account. In addition, the paper constructs four new Fed-LDL benchmarks, and proves that client-specific calibration is strictly better than uniform calibration under heterogeneous supervision quality.

**Compliance With Llm Reviewing Policy:**

Affirmed.

**Key Questions For Authors:**

1. What is the concrete release plan for the benchmarks?
2. Could the authors provide more precise implementation details for reproducibility?
3. What is the extra compute overhead from the GSA?  Since it uses the global model to calculate the logits. Have you measured how much extra time and memory this takes?

**Limitations:**

yes

**Strengths And Weaknesses:**

**Strengths**

Overall, I think this paper is technically quite solid. The problem formulation aligns well with the method design, and the experimental section is comprehensive. The paper not only reports a main results table but also includes component ablation studies, analyses under different noise intensities, varying proportions of noisy clients, label bias, and partial participation scenarios. Moreover, the ablation progression from baseline to +A to +A+B demonstrates fairly consistent improvements, which gives us reason to believe that each component indeed plays a meaningful role. Furthermore, one of the most significant contributions of this work is the proposal of a benchmark for LDL-FL, filling the gap of missing datasets in this field. Finally, combined with the theoretical analysis provided, the overall quality of this paper reaches the acceptance bar of ICML.

**Weakness**
1. The annotation process seems pretty solid overall. Still, I think the paper would feel a bit more convincing if it reported one small statistic on annotator agreement after calibration, or how much variation was still left at the end.
2. Could you report the time cost for dataset annotation in this work? I am curious about how much time it actually takes to annotate the four datasets, especially when the annotation involves individual participants.

---

> ### Author Rebuttal · Authors · 2026-03-28
>
> **Response to Reviewer 2NNQ.**
>
> We are grateful to the reviewer for the thoughtful feedback and valuable suggestions. In the following, we respond to each of the points raised.
>
> > Q1(weakness 1): I think the paper would feel a bit more convincing if it reported one small statistic on annotator agreement after calibration.
>
> Thank you for this helpful suggestion. After calibration, the raw annotations are **reasonably consistent**. We report a compact post-calibration summary below and will add this brief summary to the revised version.
>
> - **Post-calibration consistency from raw annotations.**
>   For FER-LDL and FI-LDL, we report the *average pairwise endorsement agreement*; for KADID-LDL and PIPAL-LDL, we report the *standard deviation across annotators* together with the *mean pairwise absolute score difference*. Overall, these statistics indicate reasonably good post-calibration agreement for FER and well-controlled variation for IQA.
>
> | Dataset | Statistic | Value |
> |---|---|---|
> | FER-LDL | Pairwise endorsement agreement | $0.763 \pm 0.076$ |
> | FI-LDL | Pairwise endorsement agreement | $0.828 \pm 0.077$ |
> | KADID-LDL | Std. / Pairwise abs. diff. | $0.446 \pm 0.268$ / $0.527 \pm 0.251$ |
> | PIPAL-LDL | Std. / Pairwise abs. diff. | $1.273 \pm 1.212$ / $1.171 \pm 0.932$ |
>
> > Q2(weakness 2): Could you report the time cost for dataset annotation in this work?
>
> Thank you for this question. Each formal annotation session was limited to approximately 45 minutes. The full benchmark construction process took approximately **two months**. We will add this concise summary to the revised version.
>
> > Q3(key question 1): What is the concrete release plan for the benchmarks?
>
> Thank you for this question. In the final version, we will provide a project website so that readers can directly access the released materials.  More details about release plan can be found in our response to **Reviewer Qeqz(Q2)**.
>
> > Q4(key question 2): Could the authors provide more precise implementation details for reproducibility?
>
> Thank you for this suggestion. The implementation details of our main experiments reported in Table 1 of the main text are already provided in **Appendix B.2 and B.4**, including the backbone (ResNet-18), optimizer (SGD), learning rate (0.01), batch size (16), local epochs (5), communication rounds (100), and the main FedQual hyperparameters. To further improve reproducibility, we additionally clarify here that the default main setting uses 50 clients, 20% client participation per round, a 0.5 proportion of low-quality clients, and a Dirichlet concentration of 0.5 for the non-IID partition. We will publicly release the full configuration and code upon paper acceptance.
>
> > Q5(key question 3): What is the extra compute overhead from the GSA?
>
> Thank you for this question. Conceptually, GSA uses the broadcast global model only to provide the anchor logits **$A(x)=z(x;w_g^t)$** for the client-side regularization term, rather than introducing an additional trainable branch. We therefore measured both **wall-clock time** and **peak GPU memory** by comparing the full FedQual model against the same implementation without GSA module.
>
> **(1) Wall-clock training time.**
> We report two representative benchmarks below. On FER-LDL, the full FedQual model takes 42m18s, compared with 41m03s without GSA, corresponding to an overhead of about **3.0\%**. On KADID-LDL, the measured total training time is 2h39m38s with GSA versus 2h47m05s without GSA; this indicates that the time difference is within normal run-to-run fluctuation and does not suggest a systematic slowdown from GSA.
>
> | Dataset | With GSA | Without GSA | Relative overhead |
> |---|---|---|---|
> | FER-LDL | 42m18s | 41m03s | +3.0\% |
> | KADID-LDL | 2h39m38s | 2h47m05s | negligible (within run fluctuation) |
>
> **(2) Peak GPU memory.**
> We also measured peak GPU memory. The memory overhead is minimal across all tested datasets:
>
> | Dataset | With GSA | Without GSA | Difference |
> |---|---|---|---|
> | FER-LDL | 4371 | 4225 | +146 |
> | FI-LDL | 8635 | 8623 | +12 |
> | PIPAL-LDL | 8589 | 8579 | +10 |
> | KADID-LDL | 8577 | 8595 | -18 |
>
> Overall, both the mechanism and the measurements indicate that GSA adds only a small practical overhead in time and memory.

---

> > ### Author Rebuttal · Reviewer_2NNQ · 2026-04-02
> >
> > The rebuttals have resolved my questions, and I will keep the positive score.

---

> > > ### Author Response · Authors · 2026-04-05
> > >
> > > Dear Reviewer 2NNQ,
> > >
> > > We sincerely appreciate your thoughtful evaluation and are delighted that our rebuttal has addressed your concerns.
> > >
> > > Best regards,
> > >
> > > Authors

---

### Official Review · Reviewer_Qeqz · 2026-03-23

**Soundness:** 3
**Presentation:** 3
**Significance:** 3
**Originality:** 3
**Overall Recommendation:** 4
**Confidence:** 4

**Summary:**

This paper proposes FedQual, a quality-aware federated LDL framework that addresses heterogeneous annotation reliability via client-specific calibration and reliability-aware aggregation, supported by new benchmarks and theoretical analysis. The paper is the first to study Federated Label Distribution Learning and makes a valuable contribution by constructing four benchmark datasets, advancing the development of this emerging research area.

**Compliance With Llm Reviewing Policy:**

Affirmed.

**Final Justification:**

I appreciate the author's efforts; my concerns have been resolved, and other reviewers have also provided positive feedback. I am willing to raise my score.

**Key Questions For Authors:**

Could you elaborate on some details regarding the dataset? For example, the cost and time required for annotation?

**Limitations:**

yes

**Strengths And Weaknesses:**

Strengths

	1. The proposed method addresses a key challenge in federated label distribution learning by explicitly modeling the reliability of client-side label distributions, which is a reasonable and intuitive way to handle inaccurate supervision across clients.

	2. FedQual makes a valuable contribution by constructing four benchmark datasets, which represents the first benchmarks for Federated Label Distribution Learning, and will likely facilitate future research in this area.

	3. FedQual provides theoretical insights that support the proposed design, offering a deeper understanding of why client-specific calibration is beneficial under heterogeneous supervision quality.

Weaknesses

	1. The related work section is placed at the end of the paper, which is unconventional and may hinder readability. Could the authors clarify the rationale behind this design choice?

	2. The paper introduces several new datasets, but it is unclear how they will be made publicly available and used by the community, especially if they involve privacy-sensitive data. Could the authors elaborate on data accessibility, anonymization, and potential usage constraints?

	3. The paper would benefit from more transparency regarding the dataset creation process. In particular, it would be helpful to provide details such as:
1. the annotation cost (budget),
2. the time required for data collection and labeling,
3. and the overall pipeline for constructing the label distributions.
Such information is important for reproducibility and for assessing the practicality of extending this benchmark.

---

> ### Author Rebuttal · Authors · 2026-03-28
>
> **Response to Reviewer Qeqz.**
>
> We sincerely thank the reviewer for the thoughtful comments and for recognizing the value of studying Fed-LDL as well as the contribution of the four new benchmarks. Here we respond to each point below.
>
>  > Q1(weakness 1): The related work section is placed at the end of the paper, which is unconventional and may hinder readability.
>
> Thank you for this suggestion. The detailed related-work discussion is included in the submission, but was moved to the appendix due to the **page limit** so that the main text could focus on the core technical content. This was mainly an organization choice rather than a lack of engagement with prior work. In the revised version, we will add a concise related-work paragraph earlier in the main text to enhance readability.
>
> > Q2(weakness 2): The paper introduces several new datasets, but it is unclear how they will be made publicly available and used by the community, especially if they involve privacy-sensitive data.
>
> Thank you for raising this important point. The key clarification is that our four benchmarks are **not** built from a newly collected private image corpus. Instead, they are constructed by **re-annotating four publicly available source datasets**.
> Accordingly, we will release the following benchmark artifacts for reproducibility and community use:
>
> - **Re-annotated label distributions** for each image;
> - **Instructions for obtaining the original public datasets** and pairing them with our released annotations, together with the corresponding dataset references so that future work can properly cite the original datasets;
> - **Federated partition and simulation scripts** for reproducing the heterogeneous-quality setting;
> - **Training and evaluation code** for reproducing the reported experiments;
> - **Usage constraints and licensing notes:** the original images will continue to follow the licenses and terms of the source datasets, and we will state these clearly in the release documentation.
>
> With these components, the community will be able to reconstruct the full benchmark pipeline and directly evaluate future Fed-LDL methods under the same protocol.
>
> > Q3(weakness 3): The paper would benefit from more transparency regarding the dataset creation process.
>
> Thank you for this suggestion. These practical details are already documented in the paper, although they are currently spread across the main text and appendix. Specifically, the benchmark overview is in **Section 4**, the annotation protocol is in **Appendix D.2**, the recruitment / consent forms are in **Appendix D.3--D.4**, and the label-distribution construction procedure is in Appendix **D.5**.
>
> Here we summarize the most relevant points:
>
> - **Time and cost.**
>   Each annotation session was limited to 45 minutes with mandatory breaks. We involved 20 experts in total (10 for FER and 10 for IQA), and each expert received $100 upon completing the task. The end-to-end annotation process took **approximately two months**, and the total direct annotation cost was therefore **about $2000**.
>
> - **Benchmark construction pipeline.**
>   The pipeline includes source dataset preparation, expert recruitment and screening, pilot calibration, controlled in-lab annotation, and label-distribution construction. For FER, the final label distribution is obtained by aggregating and normalizing the multi-label votes from 10 experts; for IQA, experts score six perceptual dimensions, which are then converted into soft votes and aggregated into the final quality distribution.

---

> > ### Author Rebuttal · Reviewer_Qeqz · 2026-04-02
> >
> > I appreciate the author's efforts. My concerns have been resolved, and other reviewers have also provided positive feedback. I am willing to raise my score.

---

> > > ### Author Response · Authors · 2026-04-05
> > >
> > > Dear Reviewer Qeqz,
> > >
> > > We sincerely appreciate your thoughtful evaluation and are delighted that our rebuttal has addressed your concerns.
> > >
> > > Best regards,
> > >
> > > Authors

---

### Official Review · Reviewer_rn2y · 2026-03-23

**Soundness:** 4
**Presentation:** 4
**Significance:** 4
**Originality:** 3
**Overall Recommendation:** 5
**Confidence:** 4

**Summary:**

This paper studies Federated Label Distribution Learning (Fed-LDL) under heterogeneous annotation quality across clients. It identifies a key limitation of existing federated methods, namely their reliance on sample-size-based aggregation, which fails when client supervision quality varies. To address this, the authors propose FedQual, a quality-aware framework that combines client-specific calibration (guided by a global semantic anchor) with reliability-aware aggregation based on effective information. In addition, the paper contributes the first set of Fed-LDL benchmark datasets and provides theoretical analysis supporting the advantage of quality-aware learning. Overall, the work targets an important and underexplored problem in federated learning with imperfect supervision.

**Compliance With Llm Reviewing Policy:**

Affirmed.

**Final Justification:**

The author solved my problem, I have no other questions

**Key Questions For Authors:**

see Weaknesses.

**Limitations:**

Yes

**Strengths And Weaknesses:**

Strengths

i. The paper introduces the concept of Federated Label Distribution Learning, supported by theoretical analysis. The derivations appear reasonable and provide a certain level of theoretical justification for the proposed method.

ii. The construction of the proposed datasets suggests a substantial effort, indicating that the overall workload of the paper is sufficient.



Weaknesses

i. Could the authors include a broader discussion connecting this work to existing federated learning literature, such as robust federated learning? For example, can current robust FL methods already address the issues considered in this paper, or how does the proposed setting fundamentally differ?

ii. The paper would benefit from a clear pipeline or framework diagram to improve readability, as the current method appears somewhat complex and difficult to follow.

iii. Could the authors further clarify under what conditions label distributions become inaccurate? It would also be helpful to include more related references discussing this phenomenon.

---

> ### Author Rebuttal · Authors · 2026-03-28
>
> **Response to Reviewer rn2y.**
>
> We sincerely thank the reviewer for the positive evaluation of our work and for the constructive suggestions. Below, we address each concern in turn.
>
> > Q1(weakness 1): Could the authors include a broader discussion connecting this work to existing federated learning literature.
>
> Thank you for this important question. The key distinction is that we study **Fed-LDL**, where each sample is supervised by a *distribution-valued target* $d_{m,j}\in\Delta^{C}$, rather than a *single hard label* $y_{m,j}\in\{1,\dots,C\}$. Therefore, the unreliability in our setting is not primarily discrete label corruption or arbitrary malicious updates, but **client-specific degradation in the fidelity of label distributions**.
>
> This changes the failure mode: annotation-quality heterogeneity degrades both **local optimization** where low-quality clients learn from less faithful label distributions, and **global aggregation** where sample-size-based weighting may over-trust large but low-quality clients. This coupled local--global trust issue is exactly what FedQual is designed to address.
>
> Related FL literature can be grouped below:
>
> - **Byzantine / malicious-update robustness**[1]: these methods mainly detect, filter, or downweight anomalous client updates to make aggregation robust against adversarial behavior.
>
> - **Federated learning with noisy hard labels**[2][3][4]: these methods mainly correct, filter, or reweight discrete corrupted labels under hard-label supervision.
>
> - **Robustness to biased client participation / aggregation**[5]: these methods mainly address distributional bias introduced by non-uniform client sampling or aggregation.
>
> These directions are clearly related, but they do not explicitly model the **client-specific fidelity of distribution-valued supervision**, which is the central challenge in Fed-LDL. We will add this discussion to the main text in the revised version.
>
> > Q2(weakness 2): The paper would benefit from a clear pipeline or framework diagram to improve readability.
>
> Thank you for this helpful suggestion. We agree that a clearer procedure-level summary would improve readability. In the revised version, we will add a pipeline diagram and a concise algorithm block to further improve accessibility.
>
> > Q3(weakness 3): Could the authors further clarify under what conditions label distributions become inaccurate?
>
> Thank you for this helpful comment. In practice, the observed label distribution is often constructed by **aggregating annotations from multiple annotators**[6]. Therefore, they tend to be more reliable when more annotators (especially qualified annotators) are involved, and become less reliable when annotation support is limited or annotator quality is low.
>
> Typical cases include:
>
> - **Incomplete supervision**[7][8]: only part of the target distribution is observed, so the resulting label distribution is systematically incomplete. Prior work on incomplete LDL explicitly studies how missing components of the label distribution lead to biased supervision and how the missing mass can be reconstructed.
>
> - **Insufficient annotation support**[9][10]: When the distribution is estimated from too few or low-quality annotators, it may become a poor approximation of the true target distribution. Prior work on inaccurate LDL has studied both general and instance-dependent inaccuracies in observed label distributions.
>
> - **Biased annotation procedures**[11]: When annotators or the annotation pipeline introduce systematic bias, the resulting distribution may deviate from the intended semantics even with sufficient labels. Prior work on biased-annotation LDL examines how such bias distorts label distributions and how auxiliary supervision can help correct it.
>
> In our paper, these conditions manifest as **annotation quality heterogeneity** across clients, where different clients may provide label distributions of varying fidelity for the same Fed-LDL task. We will include this discussion in the revised version.
>
> **Reference:**
>
> [1]Byzantine-Robust Learning on Heterogeneous Datasets via Bucketing, ICLR 2022
>
> [2]FedDiv: Collaborative Noise Filtering for Federated Learning with Noisy Labels, AAAI 2024
>
> [3]FedCorr: Multi-Stage Federated Learning for Label Noise Correction, CVPR 2022
>
> [4]Robust federated learning with noisy labels, IEEE Intelligent Systems 2022
>
> [5]Agnostic Federated Learning, ICML 2019
>
> [6]A Case for Soft Loss Functions, HCOMP 2020
>
> [7]Incomplete Label Distribution Learning, IJCAI 2017
>
> [8]Tail-aware reconstruction of incomplete label distributions with low-rank and sparse modeling, IEEE Trans. Circuits Syst. Video Technol., 2024
>
> [9]Inaccurate Label Distribution Learning, IEEE Trans. Circuits Syst. Video Technol. 2024
>
> [10]Instance-dependent inaccurate label distribution learning, IEEE Trans. Neural Netw. Learn. Syst., 2023
>
> [11]Label Distribution Learning with Biased Annotations Assisted by Multi-Label Learning, IJCAI 2025

---

> > ### Author Rebuttal · Reviewer_rn2y · 2026-04-01
> >
> > Thank you for your response. I have confirmed my score and have no further questions.

---

> > > ### Author Response · Authors · 2026-04-05
> > >
> > > Dear Reviewer rn2y,
> > >
> > > We sincerely appreciate your thoughtful evaluation and are delighted that our rebuttal has addressed your concerns.
> > >
> > > Best regards,
> > >
> > > Authors

---

### Decision · Program_Chairs · 2026-04-30

**Decision:**

Accept (regular)

**Comment:**

This paper addresses an important and underexplored problem in federated learning, namely label distribution learning under heterogeneous annotation quality across clients. The proposed FedQual framework is well matched to this setting: it combines quality-adaptive local calibration with reliability-aware aggregation, and is further strengthened by the introduction of new Fed-LDL benchmarks and supporting theoretical analysis. Taken together, these elements make the paper timely, technically meaningful, and potentially valuable to the community.

The reviewers’ main concerns centered on the paper’s positioning with respect to the broader robust federated learning literature, the clarity of presentation, the transparency and release plan of the benchmarks, reproducibility details, and the practical overhead introduced by the GSA module.

Overall, I find the paper technically solid, well motivated, and substantially strengthened by the rebuttal. The reviewers’ final assessments are uniformly positive, and their concerns appear to have been adequately addressed.